# RNA methyltransferase SPOUT1/CENP-32 links mitotic spindle organization with the neurodevelopmental disorder SpADMiSS

*SPOUT1/CENP-32* encodes a putative SPOUT RNA methyltransferase previously identified as a mitotic chromosome associated protein. SPOUT1/CENP-32 depletion leads to centrosome detachment from the spindle poles and chromosome misalignment. Aided by gene matching platforms, here we identify 28 individuals with neurodevelopmental delays from 21 families with bi-allelic variants in *SPOUT1/CENP-32* detected by exome/genome sequencing. Zebrafish *spout1/cenp-32* mutants show reduction in larval head size with concomitant apoptosis likely associated with altered cell cycle progression. In vivo complementation assays in zebrafish indicate that *SPOUT1/CENP-32* missense variants identified in humans are pathogenic. Crystal structure analysis of SPOUT1/CENP-32 reveals that most disease-associated missense variants are located within the catalytic domain. Additionally, SPOUT1/CENP-32 recurrent missense variants show reduced methyltransferase activity in vitro and compromised centrosome tethering to the spindle poles in human cells. Thus, *SPOUT1/CENP-32* pathogenic variants cause an autosomal recessive neurodevelopmental disorder: SpADMiSS (*SPOUT1* Associated Development delay Microcephaly Seizures Short stature) underpinned by mitotic spindle organization defects and consequent chromosome segregation errors.

Error-free chromosome segregation is crucial for cell proliferation, tissue repair and organismal growth. The process requires the formation of a mitotic spindle, a bipolar structure formed by microtubules originating from a pair of separated centrosomes in somatic animal cells. Spindle microtubules interact with kinetochores formed on the centromeric region of chromosomes to ensure proper chromosome segregation during cell division. In most animal cells, centrosomes act as major microtubule organising centres (MTOCs) and form poles of a bipolar spindle where microtubules converge[1–3]. Although higher plant cells and oocytes of many animals can assemble bipolar mitotic and meiotic spindles without centrosomes, centrosome removal in somatic animal cells typically results in mitotic delay and chromosome segregation errors[4,5].

Multiple proteins and pathways are involved in mitotic spindle organization and microtubule dynamics. Centrosomes contain a pair of cylindrical centrioles surrounded by a matrix of pericentriolar material (PCM) that promotes efficient microtubule nucleation. In addition, chromatin-mediated and microtubule-mediated microtubule nucleation pathways have also been elucidated[1,6], Several PCM proteins recruit and anchor the γ-tubulin ring complex (γ-TuRC: a key nucleator of microtubule assembly) to centrosomes and also recruit microtubule nucleation effectors. As microtubules nucleate, they form a network that ultimately becomes the mitotic spindle. Bipolar spindle assembly is essential for the accurate and timely progression of mitosis[7].

Even though all dividing cells segregate chromosomes on a mitotic spindle, pathogenic variants in genes important for centrosome and spindle function, such as *CDK5RAP2* (MIM: 608201), *PCNT* (MIM: 605925), *WDR62* (MIM: 613583), and *ASPM* (MIM: 605481), cause a spectrum of human disorders that predominantly affect brain development[8–11]. These disorders include autosomal recessive primary microcephaly, autosomal recessive microcephalic osteodysplastic

✉ e-mail: eridavis@luriechildrens.org; jeyaprakash.arulanandam@ed.ac.uk; jparul@genzentrum.lmu.de; jl5098@cumc.columbia.edu

primordial dwarfism type II (MOPD-II), and autosomal recessive Seckel syndrome[12]. Despite their phenotypic variability, all these disorders share two common clinical features, reduction in cerebral cortex size and intellectual disability. Thus, the brain is particularly susceptible to defects affecting centrosome and mitotic spindle assembly and function, which may be explained by the short time window of neurogenesis, tissue-specific isoform expression, different downstream cellular and molecular pathways, and neural-specific characteristics such as polarization[13,14]. A proposed disease mechanism is that centrosome dysfunction causes spindle disorganization, chromosome segregation errors, and/or mitotic delay, that in turn lead to exhaustion of neural progenitor cells and defective brain growth by premature differentiation and cell death[13].

SPOUT1, also known as CENP-32 or C9ORF114, contains a putative SpoU-TrmD (SPOUT) RNA methyltransferase (MTase) domain. Its role in the centrosome and mitotic spindle was first discovered by a large-scale proteomics-based bioinformatics analysis of proteins associated with mitotic chromosomes[15]. SPOUT1/CENP-32 localized to mitotic spindles and kinetochores, and its depletion by siRNA resulted in unusual centrosome detachment from the mitotic spindle poles, delayed anaphase onset, and chromosome segregation errors[15,16]. An independent large-scale CRISPR-Cas9 screen in human cells showed that SPOUT1/CENP-32 is essential for cell viability and is associated with RNA modification[17].

We report the association of bi-allelic *SPOUT1/CENP-32* variants with a complex neurodevelopmental disorder in 21 unrelated families involving 28 affected individuals. Common phenotypes in these individuals include microcephaly, seizures, intellectual disability, and varying degrees of developmental delays. These findings characterize a hitherto unreported autosomal recessive neurodevelopmental disorder (NDD); we term as SpADMiSS (*SPOUT1* Associated Development delay Microcephaly Seizures Short stature). Genetic ablation of *spout1/cenp-32* in zebrafish recapitulates phenotypes observed in humans; genetically stable mutant larvae display head size defects, augmented cell death, and altered cell cycle progression. Further, in vivo complementation assays demonstrate that missense *SPOUT1/CENP-32* changes identified in humans are pathogenic. In vitro studies demonstrate that the *SPOUT1/CENP-32* variants observed in affected individuals result in a reduction of the methyltransferase activity of the protein and compromise its function required for ensuring the tethering of centrosomes to the spindle poles in a human cell line. In summary, bi-allelic variants in *SPOUT1/CENP-32* cause an autosomal recessive neurodevelopmental disorder SpADMiSS and highlight its roles in mitotic spindle organization and brain development.

## Results

### Individuals with bi-allelic rare variants in *SPOUT1/CENP-32* display neurodevelopmental delays, seizures, microcephaly and short stature (SpADMiSS)

Through a global collaboration aided by GeneMatcher[18], we identified 28 individuals from 21 unrelated families harboring rare bi-allelic variants in *SPOUT1/CENP-32* (Supplementary Fig. 1 and Supplementary Data 1). These individuals included 16 females, 12 males, with ages ranging from 11-months to 20-years, and diverse ancestries including European, Chinese, African American/Asian, German/Malaysian, Yemeni, Afghani, Egyptian, Turkish, French, Spanish, and Arab ethnicities (Saudi Arabian, Arab Iranian, and Syrian). Of note, family B has been previously reported in a cohort of 152 families with NDDs (family ID:MR035)[19].

The *SPOUT1/CENP-32* variant spectrum in these families included 15 different missense variants, one frameshift variant, one nonsense variant and one in-frame deletion (Fig. 1A and Supplementary Data 2). The following variants were recurrent, p.N86D (2 families), p.G98S (9 families), p.R200W (3 families), p.G293S (2 families), and p.T353M (3 families). All variants were absent or rare in gnomAD. The majority of the variants occurred at amino acid residues of the protein conserved across species, and are predicted to be intolerant to variation, with 13/15 missense variants with scores >0.6 and predicted to be pathogenic by the recently described AlphaMissense predictor[20] (Fig. 1B and Supplementary Data 2). All affected individuals had bi-allelic variants with 10/21 families segregating homozygous variants whereas 11/21 pedigrees had compound heterozygous variants. Of the 10 families with homozygous variants, 7 were homozygous for the recurrent p.G98S variant. Sanger sequencing was performed for confirmation and segregation of *SPOUT1/CENP-32* variants in selected families as mandated by research or clinical workflows (Supplementary Fig. 2A).

Commonly observed clinical findings in individuals with bi-allelic *SPOUT1/CENP-32* variants included neurodevelopmental phenotypes of global developmental delay (DD) -100% (28/28), intellectual disability (ID) -100% (14/14), and seizures - 71% (20/28) (Table 1). Global development delays included motor delay, speech and language delay, and variable cognitive delays ranging from mild to severe ID. In many families, affected individuals were non-verbal and non-ambulatory. Measurements for height, weight, and head circumference were significantly lower than average in ~ 75% of affected individuals. Variable dysmorphic features were seen in 50% (14/28) individuals including high arched palate, prominent ears, upturned nostrils, tented upper lip, and high forehead (Supplementary Data 1).

Electroencephalography (EEG) tracings were reviewed in detail for 5 individuals, which demonstrated a relatively consistent electro-clinical phenotype, including disorganization in wakefulness, discontinuity in sleep, and abundant posterior-predominant sleep-potentiated spikes and polyspikes (Supplementary Fig. 2B). Around 43% (12/28) of individuals met the criteria for epileptic encephalopathy (Lennox Gastaut syndrome and infantile spams/hypsarrhythmia). Typical seizure types included infantile epileptic spasms, tonic seizures, and focal-onset seizures. Abnormal MRI findings were found in 64% (18/28) individuals, with notable findings including cerebral atrophy with enlarged ventricles, T2 hyperintensity in the white matter, and thinning of the corpus callosum (Supplementary Fig. 2C). These patients with early onset epilepsy and epileptic spasms suggest the *SPOUT1* spectrum of neurodevelopmental phenotypes include developmental and epileptic encephalopathy.

Importantly, this constellation of features in the *SPOUT1/CENP-32* cohort overlaps with phenotypes seen in other "centrosome-based diseases"[21], caused by genes involved in centrosome or spindle function including *CDK5RAP2*, *PCNT*, *WDR62*, and *ASPM*[12]. Consistent with an autosomal recessive inheritance pattern, heterozygous carrier parents in the families described in this cohort are reportedly normal and unaffected, with no neurodevelopmental phenotypes encountered as part of the clinical work-up. Mortality and stillbirths observed in the described families in this cohort appear to be comparable to those expected in the general population. However, due to the relatively small sample size of our cohort, further studies with more families will be needed in the future.

### Zebrafish *spout1/cenp-32* depletion causes neuroanatomical defects

To determine whether loss of *SPOUT1/CENP-32* is causative for phenotypes observed in affected individuals, we generated zebrafish models of *spout1/cenp-32* by ablation (CRISPR/Cas9) or suppression (morpholino, MO). We and others reported previously that zebrafish is a robust model to investigate neurodevelopmental defects in humans[22–27]. Reciprocal BLAST with the human SPOUT1/CENP-32 protein sequence (NP_057474.2) against the translated zebrafish genome identified a sole *SPOUT1/CENP-32* ortholog, encoding three *spout1/cenp-32* transcripts (canonical isoform; ENSDART00000099535.5; GRCz11; Supplementary Fig. 3A) for which the encoded protein has 76% identity and 88% similarity to human SPOUT1/CENP-32. Consistent with human expression data (GTEx; Human Protein Atlas), RNA in situ

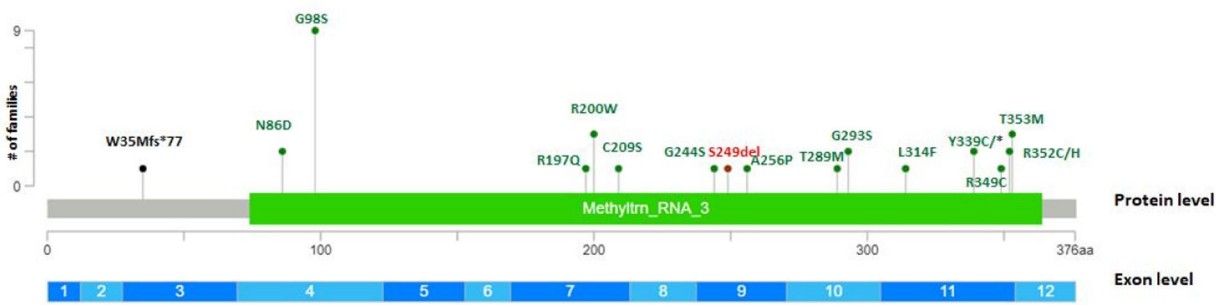

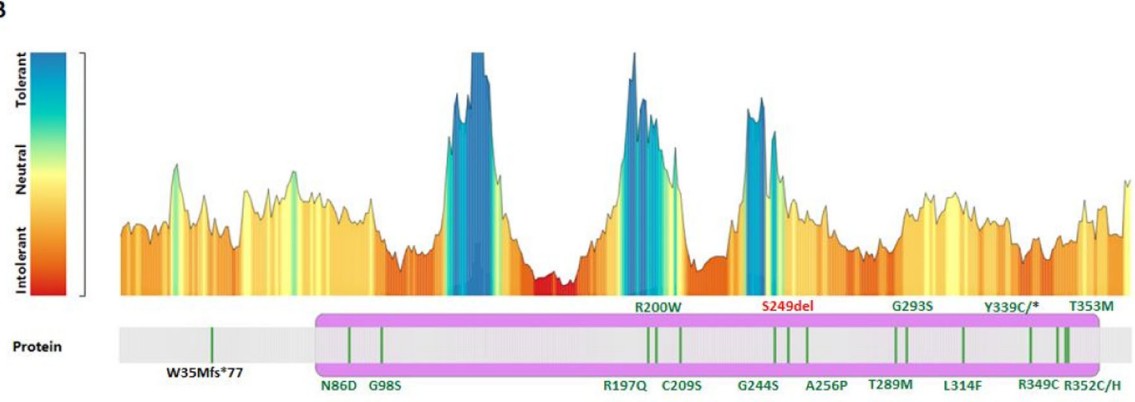

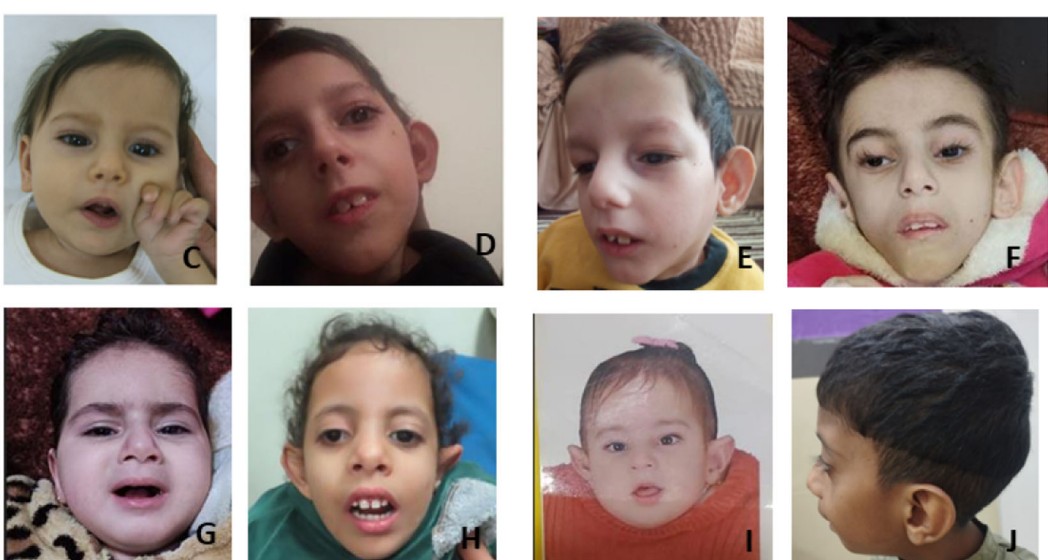

**Fig. 1 | Majority of *SPOUT1/CENP-32* variants identified in affected individuals are located in the RNA methyltransferase domain of the protein and at amino acid residues intolerant to variation. A** Schematic of *SPOUT1/CENP-32* at the exon level (NM_016390.4) and protein level (NP_057474.2) with positions of the variants identified in our cohort and the number of alleles in unrelated families identified for each variant. Missense variants are displayed in green, loss-function variants in black and the in-frame deletion in red. The single annotated domain in green is from Pfam and reflects the RNA methyltransferase domain. **B** Overlay of SPOUT1/CENP-32 variants identified in this cohort with the protein intolerance landscape of SPOUT1/CENP-32 from MetaDome. MetaDome analyses shows the mutation tolerance at each position of the protein, with red being intolerant, yellow is neutral and blue is tolerant[100]. **C–J** Variable dysmorphic features were seen in 50% (14/28) individuals including high arched palate, prominent ears, upturned nostrils, tented upper lip and high forehead. **C** Subject: family G-II-1 at 1 year. **D** Subject: family G-II-1 at 9 years. **E** Subject: family G-II-2 at 6 years. **F** Subject: family L-II-1 at 8 years. **G** Subject: family L-II-2 at 1 year 4 months. **H** Subject: family M-V-2 at 4 years. **I** Subject: family N-IV-2 at 18 months. **J** Subject: family Q-II-1 at 9 years. Created in BioRender[101].

**Table 1 | Summary of clinical features in 28 individuals with bi-allelic *SPOUT1/CENP-32* variants**

| Clinical feature | Frequency in cohort (positive individuals/individuals evaluated, %) |
|---|---|
| Developmental delay | 28/28 (100%) |
| Intellectual disability | 14/14 (100%) |
| Epilepsy<br>-Epileptic encephalopathy (Lennox Gastaut syndrome and infantile spams/hypsarrhithmia) | 20/28 (71%)<br>-12/28 (43%) |
| Poor weight gain | 19/24 (79%) |
| Short stature | 15/18 (83%) |
| Microcephaly | 18/21 (86%) |
| Variable dysmorphic features | 14/28 (50%) |
| Eye abnormalities* | 11/28 (39%) |
| Abnormal brain MRI findings | 18/28 (64%) |
| Feeding issues | 11/28 (39%) |
| Hypertonia/spasticity | 14/28 (50%) |
| Hypotonia | 10/28 (36%) |

*Eye abnormalities included microphthalmia, nystagmus, slit-like eyes, leukocoria, and cataracts. The most common features were developmental delay, intellectual disability, seizures, microcephaly, short stature, and failure to thrive. Additional findings included variable dysmorphic features, eye abnormalities, abnormal brain MRI findings, feeding issues, hypotonia, and hypertonia or spasticity. See Supplementary Data 1 for further details.

hybridization studies in zebrafish larvae showed *spout1/cenp-32* in a near ubiquitous pattern from the one-cell to pec-fin stages with a prominent expression pattern in the brain[28]. Accordingly, *spout1/cenp-32* RNAseq data from zebrafish heads report abundant *spout1/cenp-32* transcript at 3 days post-fertilization (dpf)[29], and the zebrafish expression atlas[30] reports detectable transcript up to larval day 5. Thus, *spout1/cenp-32* expression in zebrafish is spatio-temporally appropriate to model early neurodevelopmental effects.

To determine the consequences of *spout1/cenp-32* depletion, we engineered genetically stable mutants using CRISPR/Cas9 genome editing using two single guide (sg) RNAs (sgRNA1 and sgRNA3) targeting exons 5 and 7 of the canonical isoform, respectively (Supplementary Fig. 3A and Supplementary Table S1). We injected either sgRNA with or without recombinant Cas9 protein into single-cell stage embryos, and allowed them to develop until 2 dpf. Using a combined strategy of heteroduplex analysis, molecular cloning, and sequencing of PCR amplicons flanking the sgRNA target site, we estimated an average mosaicism of 81% for sgRNA1 and 76% for sgRNA3 in F0 crispants (Supplementary Figs. 3B, C). As a proxy for microcephaly and short stature, we acquired bright field lateral images of F0 zebrafish larvae at 3 dpf to measure the head size and body length. Similar to the clinical phenotypes exhibited by affected individuals with *SPOUT1/CENP-32* variants (Table 1 and Supplementary Data 1), mosaic F0 mutants displayed a significant reduction in head size area (Supplementary Figs. 4A-D).

Next, we outcrossed F0 to WT adult zebrafish and isolated animals harboring a frameshifting 1 bp insertion allele in exon 7 (c.543_544insA; p.R215Kfs*7; Fig. S3D); these F1 heterozygotes were incrossed and F2 offspring were used for subsequent phenotyping (WT denoted as '+/+'; heterozygous as '+/-'; and homozygous mutant as '-/-'). Consistent with F0 crispant data (Supplementary Fig. 4A–F), *spout1/cenp-32* stable mutants exhibited a significantly reduced head size (20% reduction, $p < 0.0001$ for *spout1/cenp-32*$^{-/-}$ vs *spout1/cenp-32*$^{+/+}$); with modest but significant reduction in body length (Supplementary Table S2 and Fig. 2A–C). We monitored mean activity of 3 dpf larvae during a 30-minute video tracking period and found significant reduction (~50%) in *spout1/cenp-32* homozygotes compared to WT siblings, with an intermediate but significant reduction in mean activity for

heterozygotes (Supplementary Fig. 5A, B). Using qPCR to monitor mRNA expression in *spout1/cenp-32* mutant heads compared to sibling larvae, we observed a ~3-fold depletion of *spout1/cenp-32* mRNA in mutants compared to WT counterparts (Fig. 2D). Notably, we observed expected Mendelian ratios, but *spout1/cenp-32* homozygous mutants die by 8 dpf ($n = 3$ independent clutches; $n = 300$ F2 larvae including all genotypes; Supplementary Fig. 5C). Together, these data support the involvement of *spout1/cenp-32* in the early development of anterior structures.

To validate the specificity of our stable mutant and F0 crispant phenotype data and to complement our observations with an independent method, we synthesized a splice-blocking morpholino (sbMO) targeting the splice donor site of *spout1/cenp-32* exon 4 in the canonical transcript (e4i4; Supplementary Fig. 3A and Supplementary Table S1). We determined the efficiency of the sbMO by RT-PCR on total RNA harvested from MO-injected embryos at 2 dpf. The MO induced retention of intron 4, resulting in a premature termination codon and frameshift (Supplementary Figs. 3E, F). We injected e4i4 MO into WT embryos at increasing doses (3 ng, 6 ng, and 9 ng), and performed live imaging for morphometric analysis using the same paradigm as stable mutants or F0 crispants. We observed a dose-dependent decrease in the head area in comparison to uninjected controls ($p < 0.0001$ vs UC; Supplementary Figs. 6A, B and Supplementary Table S2), with no differences detected between the highest *spout1/cenp-32* MO dose and sham or standard control MO injected at the same concentration (Supplementary Figs. 6C, D and Supplementary Table S2). Importantly, the head size phenotype was rescued by co-injecting WT human *SPOUT1/CENP-32* mRNA ($p < 0.0001$ vs MO; Fig. 3A, B, Supplementary Figs. 6C, D, and Supplementary Table S2). Thus, two independent approaches reveal that *spout1/cenp-32* reduction results in neuroanatomical deficits.

To probe further whether *spout1/cenp-32* suppression perturbs specific anterior structures, specifically CNS defects of the affected individuals in our cohort (microcephaly and thin corpus callosum); we painted axon tracts with anti-α-acetylated tubulin immunostaining in zebrafish larvae at 3 dpf. Quantification of optic tecta size and number of commissural axon tracts are established proxies in zebrafish for microcephaly and corpus callosum anomalies, respectively[23,25,31]. Consistent with our bright field lateral head size measurements, quantification of dorsal fluorescent signal of both optic tecta showed significant reduction in size between *spout1/cenp-32*$^{-/-}$ compared to their WT or heterozygous siblings ($p < 0.0001$ vs WT; Fig. 4A, B and Supplementary Table S2). Further, we counted fewer commissural axons crossing the midline between optic tecta for *spout1/cenp-32*$^{-/-}$ compared to WT or heterozygous siblings ($p < 0.0001$ vs UC; Fig. 4A, C, and Supplementary Table S2). CNS defects observed in *spout1/cenp-32* mutants were consistent with our transient suppression model, and co-injection of WT *SPOUT1/CENP-32* mRNA rescued both CNS abnormalities as compared to MO alone ($p < 0.0001$ vs MO; Supplementary Figs. 7A–C and Supplementary Table S2).

### *spout1/cenp-32* depletion in vivo causes increased apoptosis and prolonged cell cycle

Pathognomonic neuroanatomical defects have been associated with apoptosis and/or cell cycle arrest in the developing zebrafish larval head[22,24,27,32–34], *SPOUT1/CENP-32* depletion in human cell lines leads to centrosome detachment from the spindle pole and chromosome misalignment[15,16]. Additionally, mice lacking the overlapping *Spout1/Cenp-32* and *EndoG* loci show abnormal male germ cell apoptosis and embryonic lethality before implantation[35](MGI: 106544). Based on these data combined, we hypothesized that the decreased head size in zebrafish *spout1/cenp-32* models could be due to altered cell cycle progression with concomitant cell death.

Indeed, we observed increased cell death in *spout1/cenp-32* stable mutants compared to WT siblings ($p < 0.0001$ *spout1/cenp-32*$^{-/-}$ vs WT)

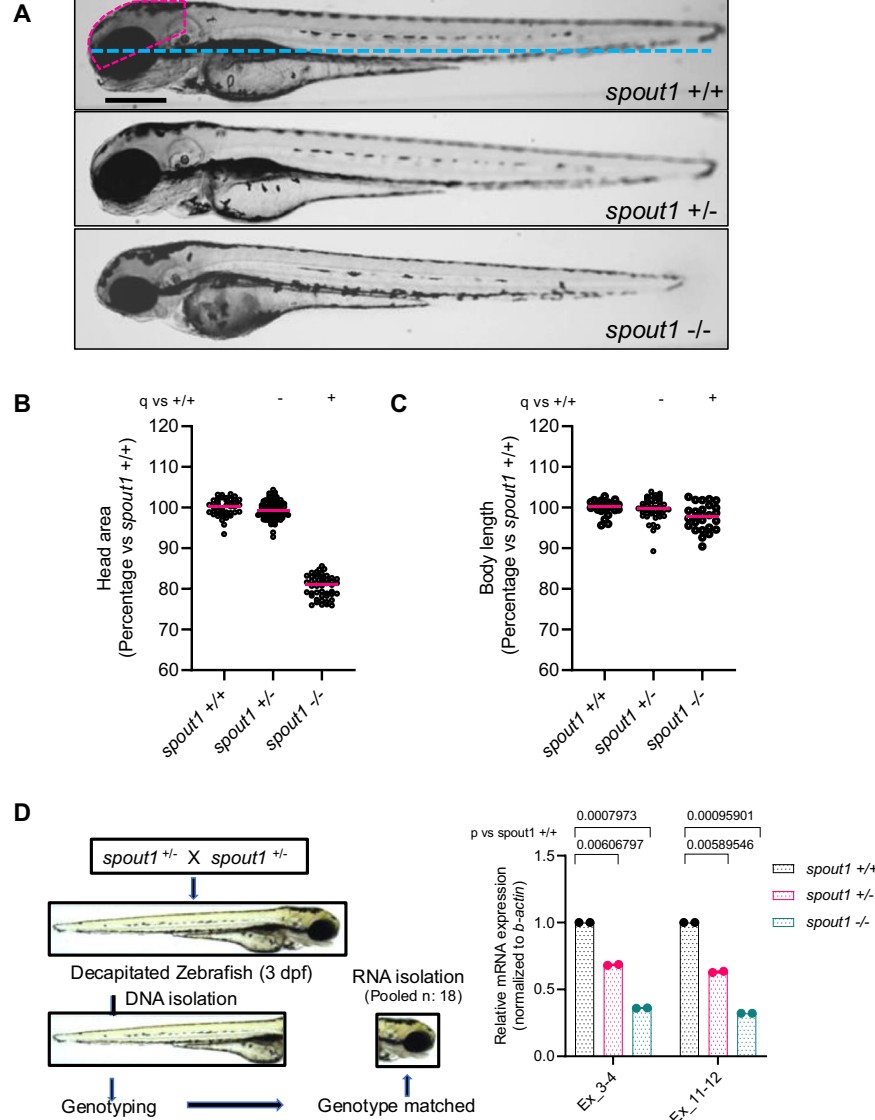

**Fig. 2 | *spout1/cenp-32* mutant larvae display reduced head area.**
**A** Representative bright field lateral images of *spout1/cenp-32*[+/+], *spout1/cenp-32*[+/-] and *spout1/cenp-32*[-/-] are shown at 3 days post-fertilization (dpf); pink dashed outline depicts head size measured, and the blue dotted line shows the body length measured. **B, C** Quantification of lateral head size (**B**) and body length (**C**) measurements were analyzed from combined experimental batches (n = 2 biological replicates). Statistical differences were calculated using a non-parametric ANOVA with Kruskal-Wallis test followed by Dunn's multiple comparisons test by

controlling False Discovery Rate (original FDR method of Benjamini and Hochberg); ( + ) and (-) indicate significant and non-significant differences, respectively. Median values are shown with pink horizontal lines. See Supplementary Table S2 for exact adjusted q-values and numbers of larvae. **D** Schematic representation (left) of sample preparation for qPCR to monitor endogenous *spout1/cenp-32* transcript (right); statistical differences were calculated using unpaired *t*-test. Two biological replicates were performed, each with technical triplicates. Scale bar, 300 μM. Source data are provided as a Source Data file.

at 2 dpf when we performed whole mount TUNEL staining (marking DNA fragmentation present in the last phase of apoptosis) and quantification of positive cells within a region of interest of the dorsal forebrain between the eyes (Fig. 4D, E and Supplementary Table S2). Monitoring cell cycle progression using whole mount pHH3 (histone H3-S10ph, a marker of G2- to M-phase transition) staining revealed a significant increase in pHH3-positive cells in a defined dorsal region of the head ($p < 0.0001$, *spout1/cenp-32*[-/-] vs WT; Fig. 4F, G and Supplementary Table S2). In independent experiments, *spout1/cenp-32* morphants showed concordant apoptosis and altered cell cycle progression defects ($p < 0.0001$ vs MO; Supplementary Figs. 7D–G and Supplementary Table S2). Together, these data support the possibility that diminished *SPOUT1/CENP-32* results in altered cell cycle progression that may result in apoptosis. In order to confirm the zebrafish findings in a mammalian neural system without triggering the

embryonic lethality preceding implantation[35], we used a transient knock-down strategy for depleting *Spout1/Cenp-32* in mice. Our results showed that down-regulation of *Spout1* in cortical stem cells in mice causes disruption of cell cycle progression (Fig. 4H, I and Supplementary Fig. 9). These results indicate that depletion of *Spout1* in neural cells leads to increased apoptosis with concomitant defects in cell cycle progression.

### SPOUT1/CENP-32 variants identified in humans are pathogenic according to zebrafish in vivo complementation assays
To bolster the evidence that the rare *SPOUT1/CENP-32* coding variants in our human cohort are pathogenic, we performed in vivo complementation assays on eleven variants[31,36,37]. We co-injected *spout1/cenp-32* sbMO with human *SPOUT1/CENP-32* mRNA harboring each disease-associated variant (p.G98S, p.T353M, p.N86D, p.G293S,

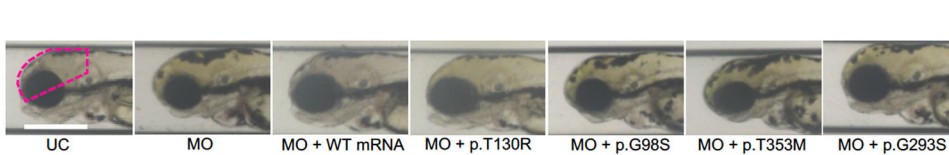

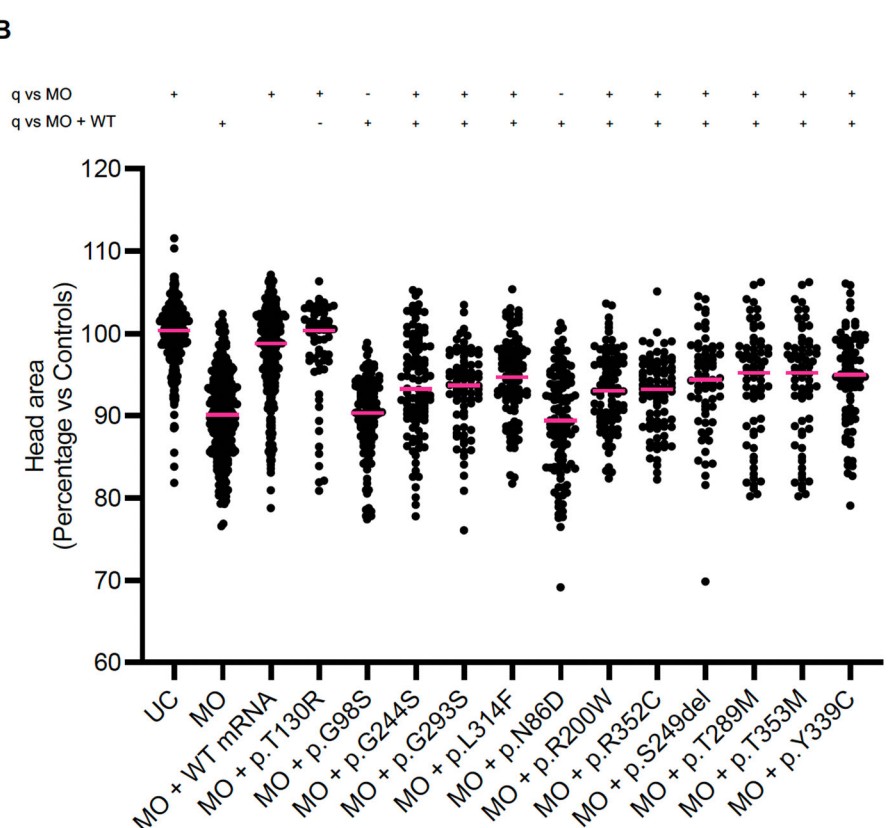

**Fig. 3 | In vivo complementation assay indicates that variants identified in affected individuals are pathogenic. A** Representative bright field images of uninjected control (UC), morphants (MO), and MO plus mRNA-injected larvae at 3 dpf. **B** In vivo complementation data show that variants identified in affected individuals are pathogenic. p.T130R, rs6478854 is a presumed benign variant used as a negative control. Scale bar, 300 μM. Statistical differences were calculated using non-parametric ANOVA with Kruskal-Wallis test followed by Dunn's multiple comparisons test by controlling False Discovery Rate (original FDR method of Benjamini and Hochberg); ( + ) and (-) indicate significant and non-significant differences, respectively. Median values are shown with pink horizontal lines. See Supplementary Table S2 for exact adjusted q-values and numbers of larvae. Source data are provided as a Source Data file.

p.R200W, p.R352C, p.G244S, p.L314F, p.Y339C, p.T289M, and p.S249del), and compared the rescue efficiency to WT human *SPOUT1/CENP-32* mRNA using head size as a phenotypic readout. We also tested p.T130R, a common variant in gnomAD, expected to be benign (dbSNP ID: rs6478854; 16,933 homozygotes of 130,923 individuals, negative control). We found that mRNA encoding disease-associated variants p.G98S or p.N86D failed to rescue the MO-induced head size phenotype, and were indistinguishable from morphants, suggesting a loss of function (Fig. 3A, B, Supplementary Fig. 11, and Supplementary Table S2). Further, we observed partial rescue of the phenotype in larvae co-injected with MO and mRNAs encoding each of p.G244S, p.G293S, p.R200W, p.R352C, p. L314F, p.S249del, p.T289M, p.T353M, and p.Y339C (Fig. 3A, B, Supplementary Fig. 11, and Supplementary Table S2). Notably, all three recurrent variants (p.G98S, p.G293S and p.T353M) scored as pathogenic and p.T130R scored as benign, supporting the sensitivity and specificity of our assay. We detected modest differences ( ≤ 3%) in body length for some variants in the in vivo complementation assays (Supplementary Figs. 10A, B, Supplementary Fig. 11, and Supplementary Table S2), but the physiological relevance

of these observations is unclear given the small dynamic range of the MO phenotype. Additionally, ectopic expression of WT or mutant *SPOUT1/CENP-32* mRNA in the absence of sbMO was indistinguishable from un-injected controls (Supplementary Figs. 12A, B and Supplementary Table S2).

Together, our zebrafish modeling studies support a critical role for *SPOUT1/CENP-32* in neurodevelopment and confirm the pathogenicity of eleven of the missense variants in our cohort.

**Structural analysis of SPOUT1/CENP-32 reveals a SAM-dependent SPOUT methyltransferase domain suitable for catalytic activity**

In order to determine the molecular mechanism for the pathogenicity of the variants seen in our patient cohort and confirmed by in vivo studies using zebrafish, we characterized the structure of the SPOUT1/CENP-32 protein (Fig. 5). This protein has a predicted SPOUT domain with an oligonucleotide binding (OB) fold inserted into the catalytic domain (Fig. 5A and Supplementary Fig. 13A). It also contains a highly basic N-terminal region predicted to form an α-helix (amino acid

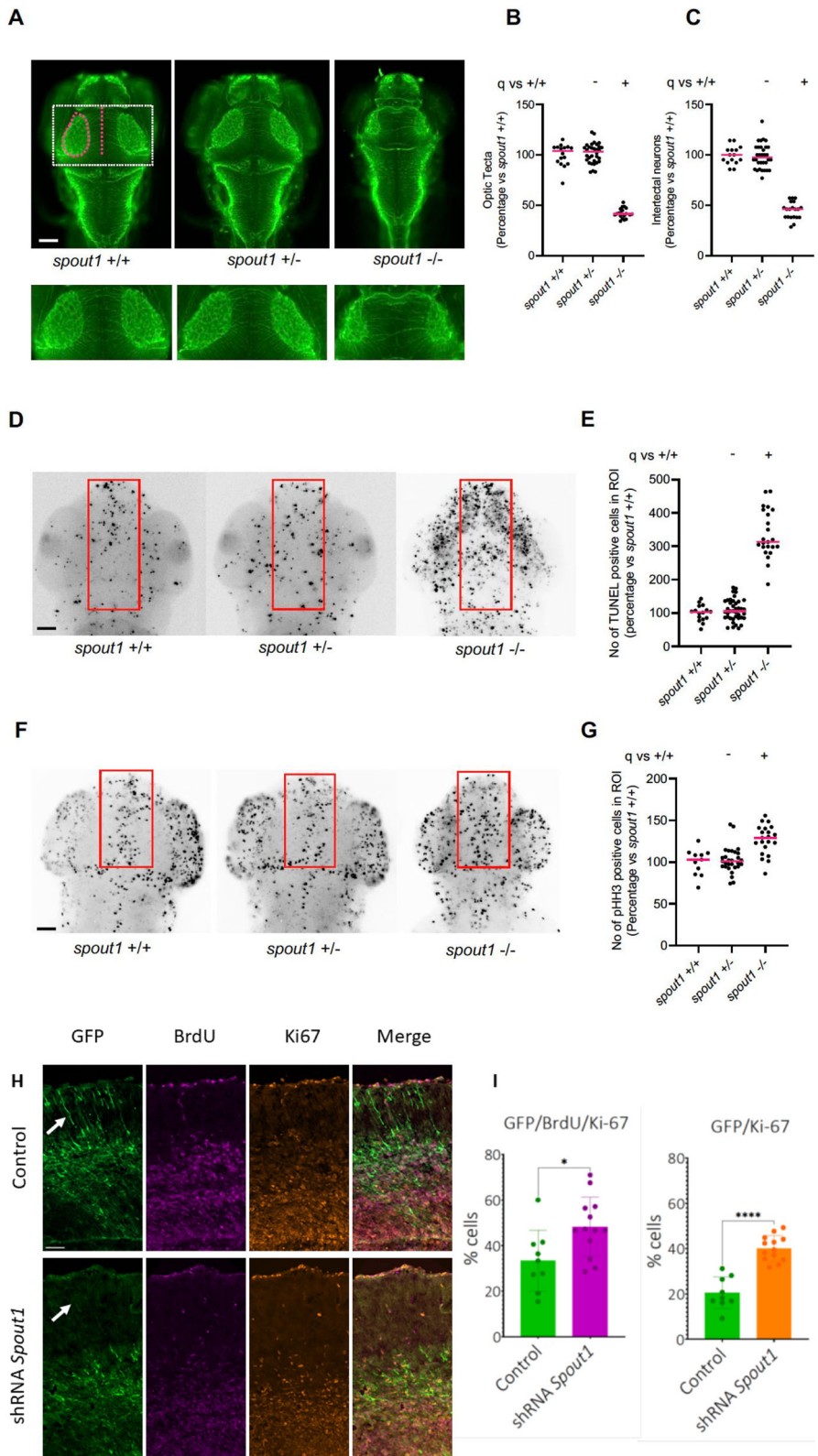

residues 20-64; pI = 9.48; Fig. 5A and Supplementary Fig. 13A). To gain insights into SPOUT1/CENP-32 structure and function, we crystallized the predicted methyltransferase (MTase) domain (SPOUT1/CENP-32 71-376) made recombinantly in *E. coli* (Supplementary Fig. 13B). The crystal structure was determined to a resolution of 2.38 Å using Single Anomalous Dispersion (SAD) phasing using Seleno (Se)-methionine labeled SPOUT1/CENP-32 (Fig. 5B and Supplementary Table S3).

Structural analysis confirmed that SPOUT1/CENP-32 is a SPOUT MTase with a trefoil knot fold consisting of parallel six stranded β-sheets sandwiched between two layers of α-helices (Fig. 5B) that promote dimerization of the protein. A common feature of SPOUT MTases is the presence of additional domains fused to their N- or C-terminus or inserted between domains within the SPOUT MTase. The SPOUT1/CENP-32 OB-fold[38–40] is inserted into the SPOUT MTase domain (amino

**Fig. 4 | *spout1/cenp-32*<sup>-/-</sup> mutant larvae have CNS defects, apoptosis and altered cell cycle progression. A** Representative dorsal images of wholemount larvae fixed at 3 dpf and immunostained for acetylated tubulin to demarcate axon tracts. We counted commissural axons crossing the midline between the optic tecta (pink dashed line); and the area of optic tecta (pink dashed oval). Dashed white box (upper panel) represents magnified image in lower panel. Scale bar, 100 μM. **B**, **C** Quantification of optic tecta size and intertectal neuron number, respectively. **D** Representative dorsal inverted fluorescent images of zebrafish larvae marked by TUNEL at 2 dpf. The region of interest (ROI) quantified is shown with a red rectangle. Scale bar, 100 μM. **E** Quantification of TUNEL stained cells as measured in the ROI. **F** Representative dorsal inverted fluorescent images showing phospho-histone H3 (pHH3) positive cells at 2 dpf. The ROI quantified is shown with a red rectangle. Scale bar, 100 μM. **G** Quantification of pHH3 positive cells as measured in ROI. Data shown in **B**, **C**, **E**, and **G** are combined from two biological replicates. Statistical differences were calculated using non-parametric ANOVA with Kruskal-Wallis test followed by Dunn's multiple comparisons test by controlling False Discovery Rate (original FDR method of Benjamini and Hochberg); (+) and (·) indicate significant and non-significant differences, respectively. Median values are shown with pink horizontal lines. See Supplementary Table S2 for exact adjusted q-values and numbers of larvae. **H**, **I** Murine cortical progenitor cells electroporated with shRNA against Spout1 less frequently leave mitotic cell cycle both within 24 hours (**I**, left diagram), and 48 hours (**I**, right diagram) after electroporation, and also less frequently migrate to the cortical plate (CP) (white arrows) compared to wild type cells (Note: GFP positive neurons in the control CP and lack of them in shRNA electroporated brains). In utero electroporation of targeting constructs into the lateral ventricles was carried out at E13.5. 24 hours later, neocortical progenitors were labeled with BrdU during S phase of the mitotic cycle. Neocortical cells that express both mitotic marker Ki67 and GFP represent a fraction of the cortical cells that still proliferate 48 hours after in utero electroporation, while cells that express BrdU, Ki67 and GFP represent a fraction of cells that proliferate 24 hours after BrdU injection. Statistical differences were calculated using two-sided unpaired t-test (*, $P \le 0.05$; ****, $P \le 0.0001$). **I** left diagram $P = 0.0176$. Data are presented as Mean +/- SD. Number of brain samples analyzed: 3 (control), 4 (experiment); number of slices analyzed: 9 (control), 13 (experiment). Scale bar, 50 μM. Source data are provided as a Source Data file. Created in BioRender[102].

acid residues 181 to 257) (Fig. 5B and Supplementary Fig. 13A). Like the OB-fold in SPOUT1/CENP-32, most of these additional domains in other SPOUT MTases are involved in nucleic acid recognition, suggesting that SPOUT methyltransferases are mostly involved in nucleic acid binding and methylation[41]. Notably, SPOUT1/CENP-32 is the only SPOUT superfamily member with the OB-fold insertion within the catalytic domain[41].

SPOUT MTases use S-adenosyl-L-methionine (SAM) as the methyl group donor for the methyltransferase reaction. The products of this reaction are the methylated substrate and S-adenosyl-homocysteine (SAH)[41,42]. Although we did not supplement SPOUT1/CENP-32 with either SAM or SAH during protein purification or crystallization, our crystal structure showed SAH bound within the cofactor binding site (Fig. 5B–D). This strongly suggests that SPOUT1/CENP-32 is an active MTase that stably retains the SAH by-product of its methyltransferase reaction (Supplementary Fig. 14A and Supplementary Figs. 14D, E).

The cofactor binding site is a deep pocket formed between the six-stranded beta-sheet and the helices involved in dimerization. SAH is stabilized via extensive hydrophobic and hydrogen bonding interactions involving the adenine and ribose moieties of the SAH and catalytic site residues (Fig. 5D). To determine whether SPOUT1/CENP-32 can also bind the cofactor, SAM, and to gain further structural insights into the substrate binding site, we soaked the SPOUT1/CENP-32 crystals in a cryo-protectant solution containing a molar excess of SAM prior to freezing the crystals for X-ray data collection. These crystals diffracted X-rays beyond 2.62 Å (Fig. 5E, F and Supplementary Fig. 14F). The SAM-bound structure of SPOUT1/CENP-32 was determined using molecular replacement. The difference electron density map calculated for the refined model showed a well-defined electron density for the bound SAM (Supplementary Figs. 14B, D, E). The mode of SAM binding is similar to that of SAH, with a methyl group pointing out of the deep cofactor binding pocket (Fig. 5C–F).

Our SPOUT1/CENP-32 structures also reveal that SPOUT1/CENP-32 dimerizes via a helical segment in the catalytic domain that packs almost perpendicularly against an equivalent segment from its dimeric counterpart (Fig. 5B and Supplementary Fig. 14F). This mode of dimerization stabilizes the conformation of the substrate binding pocket and hence is likely crucial for the enzymatic activity. To ensure that SPOUT1/CENP-32 dimerization is not an artifact of crystal packing and to test whether SPOUT1/CENP-32 is a homodimer in solution, we performed size exclusion chromatography combined with multi-angle light scattering (SEC-MALS). The molecular weight measured via SEC-MALS is 66.06 ± 1.26 kDa, which matches with a calculated molecular weight for a homodimeric assembly (67 kDa) (Supplementary Fig. 14G). This is consistent with reports that all studied SPOUT family members are known to form dimers, with dimerization suggested to be essential for MTase activity[41].

## SPOUT1/CENP-32 is an active RNA MTase

SPOUT MTases are the second largest family of RNA MTases, with most members involved in RNA metabolism and ribosome biosynthesis[43]. The crystal structure suggests that SPOUT1/CENP-32 is likely an active enzyme, so we evaluated its activity by performing in vitro MTase-GloTM assays (Promega)[44] using recombinant SPOUT1/CENP-32. Recombinant NSUN6, a well characterized MTase[45] and Survivin, a protein with no nucleic acid binding activity, were used as positive and negative controls, respectively (Fig. 6A). The bioluminescence signal obtained in the MTase assay directly corresponds to the amount of SAH produced in the reaction. Our results reveal that SPOUT1/CENP-32 has an intrinsic activity that is comparable to the MTase activity of NSUN6 for total RNA extract (Fig. 6A). As expected, Survivin showed no MTase activity (Fig. 6A). This demonstrates that SPOUT1/CENP-32 retains enzymatic activity towards RNA.

To assess if the observed activity is specific and depends on SPOUT1/CENP-32's ability to bind its cofactor SAM, we mutated Ala356, an amino acid residue located at the centre of the SAM binding pocket, to Asn (an amino acid residue with a relatively longer side chain; A356N) and characterized this variant by determining its crystal structure and by subjecting it in our MTase activity assay (Supplementary Fig. 14C and Fig. 6B). The crystal structure of SPOUT1/CENP-32 A356N determined at 2.5 Å resolution, showed that while this variant does not affect the overall structure or the conformation of SAM binding pocket, the variant protein failed to stably bind SAH in the expression host (*E. coli*), confirming the perturbation of cofactor binding (Supplementary Figs. 14A–C). We tested the SPOUT1/CENP-32 A356N variant in our MTase activity assays and observed that this variant shows reduced activity, indicating that the specific catalytic activity of SPOUT1/CENP-32 depends on cofactor binding (Fig. 6B and Supplementary Fig. 14H).

## Catalytic activity of SPOUT1/CENP-32 is necessary for centrosome tethering to the spindle poles

Consistent with previous studies[15,16], depletion of SPOUT1/CENP-32 using siRNA oligos results in cells showing a unique centrosome detachment phenotype, with more than 80% of the cells showing centrosomes detached from spindle poles of bipolar spindles and often migrating towards the metaphase plate (Fig. 6C, D and Supplementary Figs. 15A, B). Expression of wild type (WT) SPOUT1/CENP-32-GFP in U2OS stable cell lines significantly rescues this phenotype when endogenous SPOUT1/CENP-32 is depleted (Fig. 6C, D and Supplementary Figs. 15C–F).

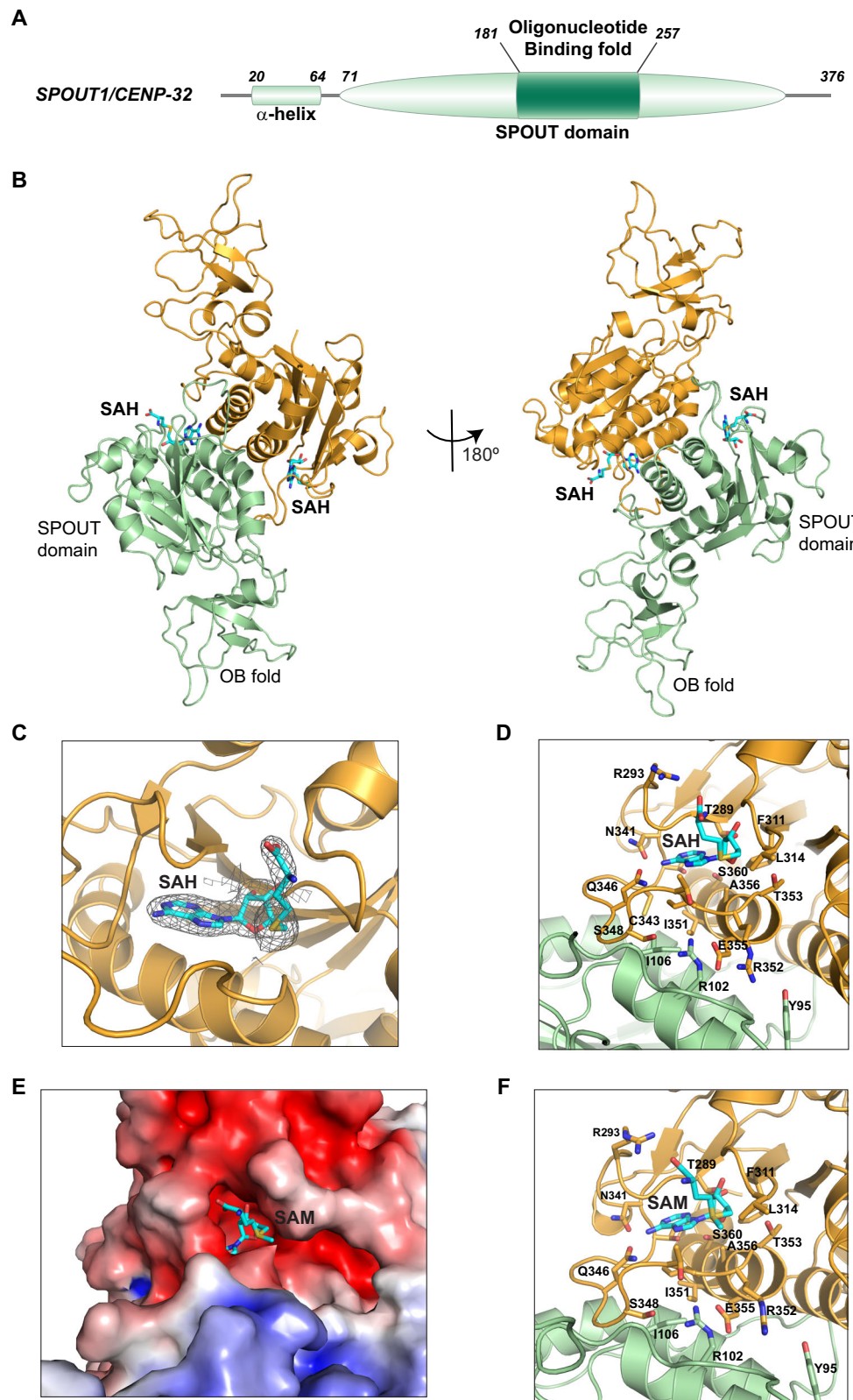

**Fig. 5 | Structural characterization reveals that SPOUT1/CENP-32 is a SPOUT Methyltransferase that dimerizes through its catalytic domain. A** Domain architecture of SPOUT1/CENP-32 with the structural domains of SPOUT1/CENP-32 highlighted. **B** Cartoon representation of the crystal structure of SPOUT1/CENP-32 71-376 bound to S-adenosyl homocysteine (SAH) in two orientations (left and right panel). The main domains of SPOUT1/CENP-32 (SPOUT domain and OB-fold) are highlighted. SAH is bound to the cofactor pocket, indicating that SPOUT1/CENP-32 could be an active methyltransferase. Our structural analysis is consistent with the structure that was deposited by SGC while this manuscript was in preparation (PDB: 4RG1). **C**, **D** and **F** Close-up of the active site of SPOUT1/CENP-32 with SAH (**C** and **D**) and with SAM (**F**) showing the electron density map for SAH (**C**) and the amino acid residues that are responsible for the interactions that stabilize cofactor binding. The binding of SAH and SAM is similar. **E** Close-up of the electrostatic surface potential of SPOUT1/CENP-32 71-376 bound to SAM showing the deep pocket formed by the trefoil knot.

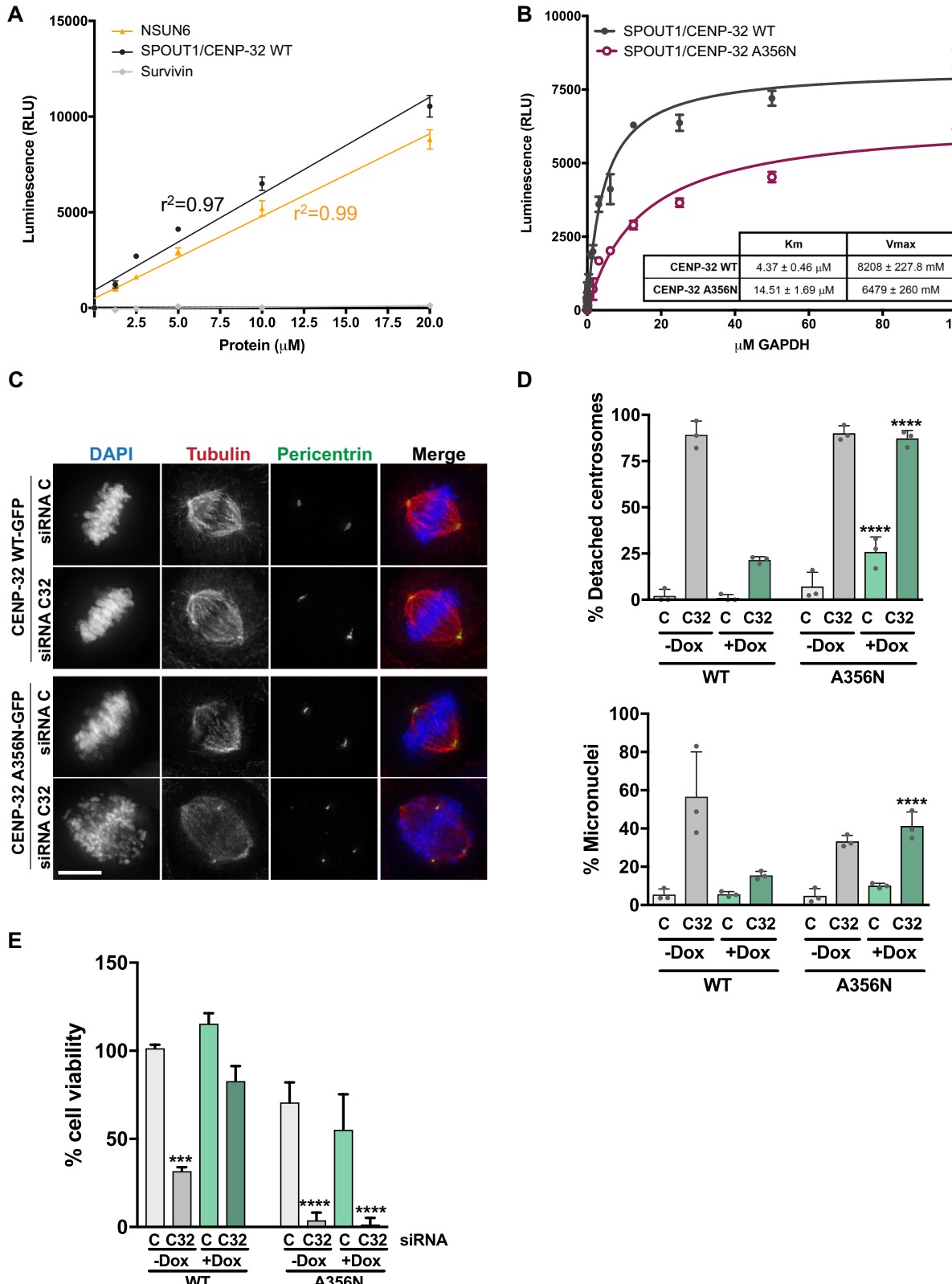

Incorrectly segregated chromosomes often result in the formation of micronuclei. SPOUT1/CENP-32 depletion leads to an increase in the percentage of micronucleus formation that can be rescued with induction of SPOUT1/CENP-32 WT-GFP expression, indicating that SPOUT1/CENP-32 is necessary for accurate chromosome segregation (Fig. 6D). We tested the A356N catalytic variant in our siRNA rescue assays to check whether the methyltransferase activity of SPOUT1/CENP-32 is necessary for its function. Depletion of SPOUT1/CENP-32 and induction of SPOUT1/CENP32 A356N-GFP expression in our stable cell lines failed to rescue the centrosome detachment and micronucleus formation phenotype (Fig. 6C, D and Supplementary Figs. 15A–F). Thus, SPOUT1/CENP-32 methyltransferase activity is required for maintaining spindle integrity (Fig. 6D).

**Fig. 6 | SPOUT1/CENP-32 is an active Methyltransferase and its activity is crucial for its mitotic function. A** Titration of methyltransferases (SPOUT1/CENP-32 and NSUN6) to assess the methyltransferase activity of SPOUT1/CENP-32 in vitro where Survivin protein was used as a negative control. Increasing amounts of either methyltransferase (NSUN6, a well-known RNA methyltransferase, or SPOUT1/CENP-32) or Survivin protein (0–20 μM) were incubated with 1 μg of total RNA extract (isolated from NSUN6 depleted or SPOUT1/CENP-32 depleted U2OS cells) and 20 μM SAM for 30 min at room temperature. RLU (Relative Luminescence unit). Data from three biological replicates, $n = 9$. Data represented as mean ± SEM. Source data are provided as a Source Data file. **B** Methyltransferase assay to determine the Km and Vmax of SPOUT1/CENP-32 WT and A356N in the presence of constant SAM (20 μM) and increasing amounts of a *GAPDH* mRNA hairpin as substrate (GCCCCCUCUGCUGAUGCCCCCAUGUUCGUCAUGGGUGUGAA; 0 – 100 μM). Km and Vmax values for SPOUT1/CENP-32 proteins are depicted in the table. RLU (Relative Luminescence unit). Data from three biological replicates, $n = 6$. Data represented as mean ± SEM. Source data are provided as a Source Data file. **C** Representative immunofluorescence images of SPOUT1/CENP-32 inducible U2OS cell lines for the analysis of centrosome detachment upon Control (siRNA C) or SPOUT1/CENP-32 (siRNA C32) depletion using siRNA oligos and rescue with either induction of SPOUT1/CENP32 WT-GFP or SPOUT1/CENP32 A356N-GFP expression. Conditions with doxycycline only. Conditions without doxycycline in Fig. S15F.

**D** Quantification for the analysis of the centrosome detachment phenotype (% of cells with detached centrosomes; top panel) and chromosome segregation errors (% of micronuclei; bottom panel) in inducible U2OS cell lines expressing SPOUT1/CENP-32 WT-GFP or SPOUT1/CENP-32 A356N-GFP. Data are representative of a minimum of three biological replicates, mean ± SD, $n = 3$. Scale bar, 10 μm; Two-sided Fisher's exact test with Bonferroni correction (****, $P \le 0.0001$). Exact p-values for the detached centrosome phenotype vs the WT condition are: A356N siRNA C +Dox=2.42E-06, A356N siRNA C32 +Dox=7.90E-20. Exact p-values for the micronuclei phenotype vs the WT condition are: A356N siRNA C32 +Dox=1.43E-11. Data points for each of the replicates are shown. Source data are provided as a Source Data file. **E** Cell viability evaluation of inducible U2OS cell lines expressing SPOUT1/CENP-32 WT-GFP or SPOUT1/CENP-32 A356N-GFP assessed by the MTT assay. Three independent biological replicates, mean ± SEM. WT siRNA C -Dox: $n = 26$; WT siRNA C32 -Dox: $n = 26$; WT siRNA C +Dox: $n = 25$; WT siRNA C32 +Dox: $n = 27$; A356N siRNA C -Dox: $n = 11$; A356N siRNA C32 -Dox: $n = 18$; A356N siRNA C +Dox: $n = 13$; A356N siRNA C32 +Dox: $n = 18$. Non-parametric ANOVA with Kruskal-Wallis followed by Dunn's multiple comparisons test (***, $P \le 0.001$; ****, $P \le 0.0001$). Exact p-values vs the WT siRNA C -Dox condition are: WT siRNA C32 -Dox=1.44E-04, A356N siRNA C32 -Dox=1E-06, A356N siRNA C32 +Dox=1E-06. Source data are provided as a Source Data file.

We also assessed the effect of SPOUT1/CENP-32 depletion on cell proliferation using the MTT (3-(4,5-Dimethylthiazol-2-yl)-2, 5-Diphenyltetrazolium bromide) colorimetric assay. Consistent with our zebrafish cell death data (Fig. 4D, E and Supplementary Figs. 7D, E), SPOUT1/CENP32 depletion in U2OS cells led to a decrease in cell viability (31.7 ± 2.3 % for siRNA C32 vs 101.5 ± 1.9 % for siRNA C; Fig. 6E). As expected, we also observed reduced cell viability for cells expressing SPOUT1/CENP32 catalytic variant (SPOUT1/CENP32 A356N-GFP) with reduced methyltransferase activity (Fig. 6E). These data together indicate that the role of SPOUT1/CENP-32 in regulating spindle organization is dependent on its methyltransferase activity.

### Disease-associated SPOUT1/CENP-32 missense variants show reduced methyltransferase activity

To understand the effect of the SPOUT1/CENP-32 patient variants on the activity and function of SPOUT1/CENP-32, we first mapped the amino acid variations onto the SPOUT1/CENP-32 crystal structure (Fig. 7A). Several of these variants are clustered around the cofactor binding pocket (N86D, T289M, G293S, L314F, Y339C, R352C and T353M) while others are present at the dimeric interface (G98S) and within the OB-fold (R200W, G244S and S249del) (Fig. 7A). Considering the proximity of the variants to the catalytic site and the nature of the amino acid changes, and recurrence in affected individuals, we assessed the effect of N86D, G98S, T289M, G293S and T353M missense variants on the catalytic activity of SPOUT1/CENP-32. We also assessed the G244S missense variant as this change is located in the OB-fold and might affect substrate binding. We used SPOUT1/CENP-32 T130R, a common variant expected to be benign and to behave like the WT protein, as a negative control.

We assessed the MTase activity of the selected SPOUT1/CENP-32 patient variants in our in vitro MTase activity assays. It has been shown previously that SPOUT1/CENP-32 recognizes secondary structures of RNA, specifically to the RNA stem[46]. We used the hairpin formed by a 41nt sequence of the *GAPDH* mRNA as a substrate and indeed, including this substrate in the assay stimulated the catalytic activity of SPOUT1/CENP-32 WT (Fig. 7B). The SPOUT1/CENP-32 variants seen in patients show a 2-fold increase in Km and/or a decrease in the Vmax, while T130R showed comparable Km and Vmax to SPOUT1/CENP-32 WT (Fig. 7B and Supplementary Fig. 14H). Size exclusion chromatography and mass photometry experiments show that all variants except SPOUT1/CENP-32 G98S exist as dimers (Supplementary Fig. 16A–F). Considering that the loops forming the cofactor binding site are stabilized by the dimeric interface and all variants except G98S

are either near the cofactor binding site or within the OB-fold domain implicated in nucleic acid binding, the cofactor and/or substrate binding and turnover of the SPOUT1/CENP-32 variants may be affected, making them less efficient enzymes in affected individuals.

### SPOUT1/CENP-32 missense variants affect the role of SPOUT1/CENP-32 in maintaining spindle integrity

To further understand whether the decrease in MTase activity observed for SPOUT1/CENP-32 variants is linked to the role of SPOUT1/CENP32 in ensuring proper spindle organization, we performed siRNA rescue assays (Fig. 8 and Supplementary Fig. 17A). SPOUT1/CENP-32 was depleted in our U2OS stable cell lines using siRNA oligos targeting the endogenous mRNA and replaced with SPOUT1/CENP-32-GFP WT or SPOUT1/CENP-32-GFP variants (SPOUT1/CENP-32-GFP N86D, SPOUT1/CENP-32-GFP G98S, SPOUT1/CENP-32-GFP G244S, SPOUT1/CENP-32-GFP T289M, SPOUT1/CENP-32-GFP G293S, SPOUT1/CENP-32-GFP T353M) (Supplementary Figs. 15A–E). Consistent with the reduced activity observed for the patient variants in the in vitro MTase assays and zebrafish rescue experiments, induction of expression of the SPOUT1/CENP32-GFP variants failed to rescue both the detached centrosome phenotype and the appearance of micronuclei (Fig. 8 and Supplementary Figs. 15C–E). The T353M variant, which affects one of the key amino acid residues coordinating cofactor binding, showed the strongest phenotype of all the variants in our siRNA rescue assays with aberrant spindle abnormalities and centrosome splitting when it was expressed in endogenous SPOUT1/CENP-32 depleted cells (Fig. 8). Expression of T353M led to an increased number of cells with detached centrosomes and an increased number of micronuclei even in the siRNA control condition (Fig. 8). Similarly, when the effect of the SPOUT1/CENP-32-GFP variants on cell proliferation was assessed using the MTT colorimetric assay we observed that all variants failed to rescue the reduction on cell viability induced by SPOUT1/CENP-32 depletion, and the T353M was the variant that showed the strongest effect (Fig. 8D).

Overall, our characterization of SPOUT1/CENP-32 patient variants in vitro MTase assays and in functional rescue assays in a human cell line demonstrates that these patient-derived variants directly affect SPOUT1/CENP-32 MTase activity and compromise its role in maintaining the integrity and function of the mitotic spindle.

## Discussion

Collectively, the spectrum of disorders caused by disruption of centrosome/spindle related proteins have been referred to as

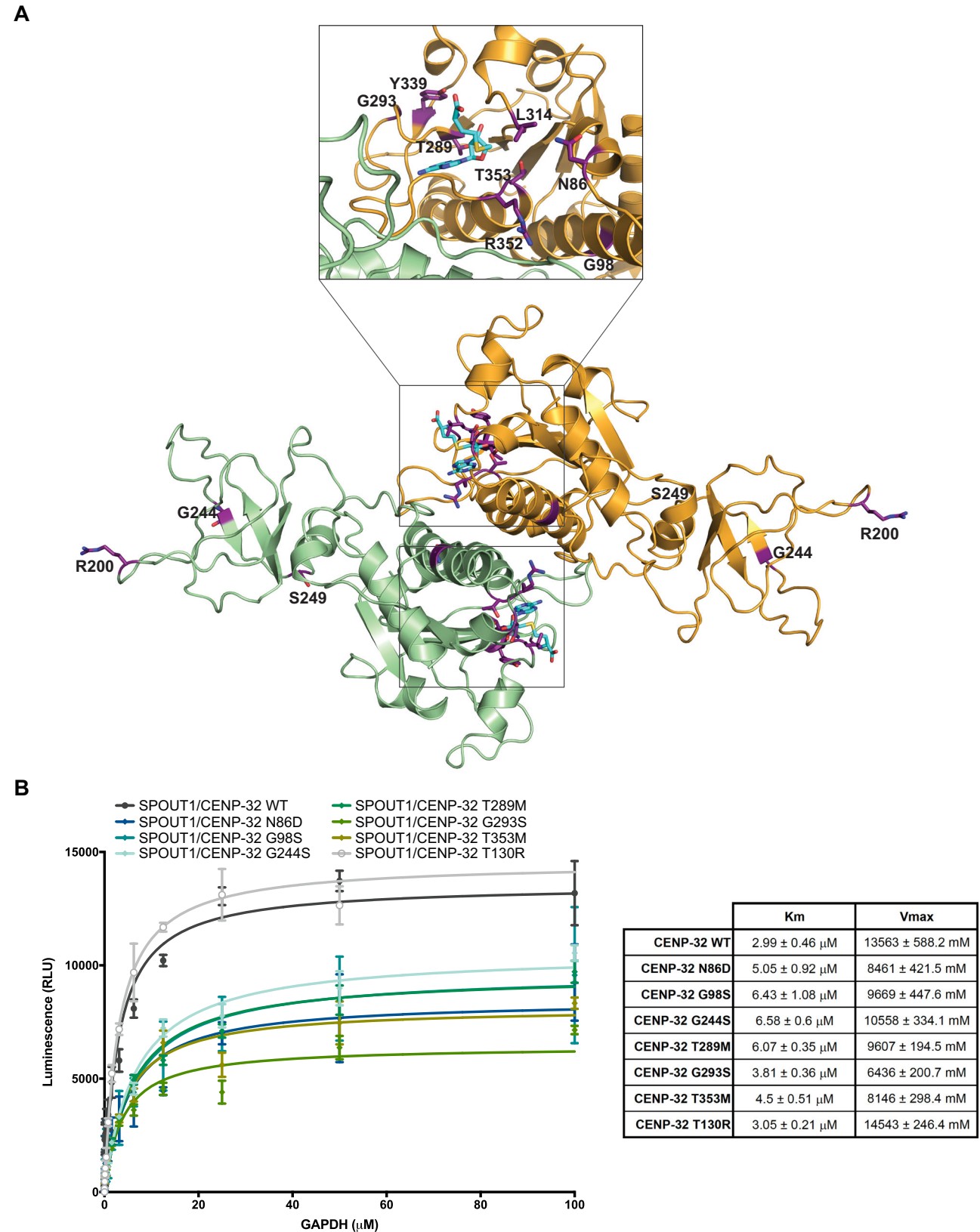

| | Km | Vmax |
|---|---|---|
| **CENP-32 WT** | 2.99 ± 0.46 μM | 13563 ± 588.2 mM |
| **CENP-32 N86D** | 5.05 ± 0.92 μM | 8461 ± 421.5 mM |
| **CENP-32 G98S** | 6.43 ± 1.08 μM | 9669 ± 447.6 mM |
| **CENP-32 G244S** | 6.58 ± 0.6 μM | 10558 ± 334.1 mM |
| **CENP-32 T289M** | 6.07 ± 0.35 μM | 9607 ± 194.5 mM |
| **CENP-32 G293S** | 3.81 ± 0.36 μM | 6436 ± 200.7 mM |
| **CENP-32 T353M** | 4.5 ± 0.51 μM | 8146 ± 298.4 mM |
| **CENP-32 T130R** | 3.05 ± 0.21 μM | 14543 ± 246.4 mM |

**Fig. 7 | SPOUT1/CENP-32 patient variants show decreased methyltransferase activity. A** Mapping of the SPOUT1/CENP-32 variants in the SPOUT1/CENP-32 71-376 SAH-bound structure. Variant residues are highlighted in purple.
**B** Methyltransferase assay to determine the Km and Vmax of SPOUT1/CENP-32 WT and the variants in the presence of constant SAM (20 μM) and increasing amounts of a *GAPDH* mRNA hairpin as substrate (GCCCCCUCUGCUGAUGCCCCCAU-GUUCGUCAUGGGUGUGAA; 0 – 100 μM; left panel). Km and Vmax values for all

SPOUT1/CENP-32 proteins are depicted in the table (right panel). R square values are as follows: SPOUT1/CENP-32 WT – 0.79 (n = 10), SPOUT1/CENP-32 N86D – 0.83 (n = 6), SPOUT1/CENP32 G98S – 0.87 (n = 6), SPOUT1/CENP-32 G244S – 0.93 (n = 10), SPOUT1/CENP-32 T289M – 0.97 (n = 10), SPOUT1/CENP-32 G293S – 0.93 (n = 10), SPOUT1/CENP-32 T353M – 0.9 (n = 10), SPOUT1/CENP-32 T130R – 0.98 (n = 4). Data from three independent biological replicates, mean ± SD. Source data are provided as a Source Data file.

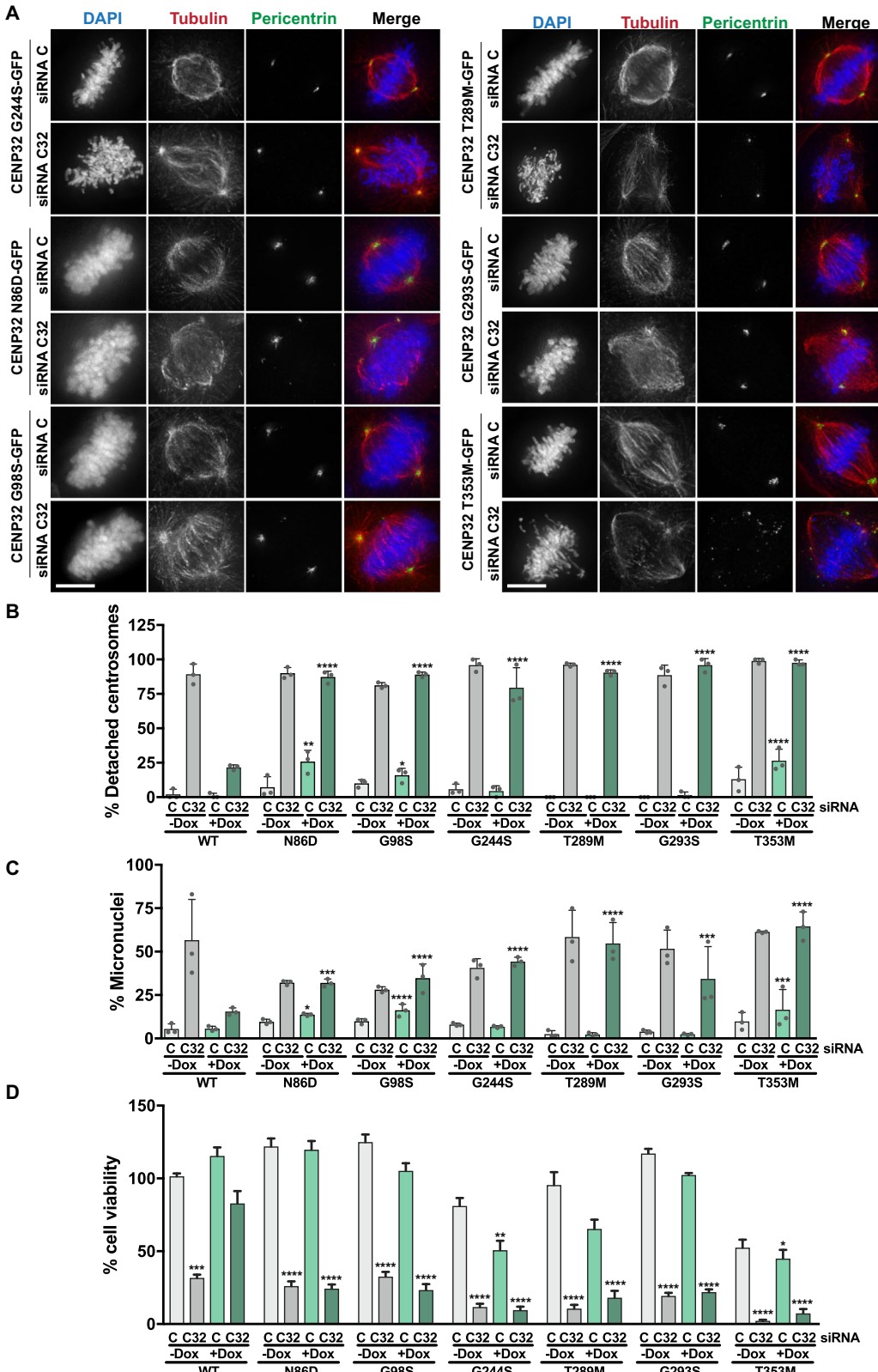

"centrosome-based diseases"[21] and the neurodevelopmental disorder associated with *SPOUT1/CENP-32* variants described in this study, adds to this group of disorders. Most centrosome-based diseases are caused by pathogenic null variants in centrosomal genes such as *ASPM, WDR62, PCNT,* and *CDK5RAP2*[47–49]. *SPOUT1/CENP-32* differs, because most of the variants reported in this study are missense or in-frame variants with decreased catalytic activity. The zebrafish

*spout1/cenp-32* mutants are lethal by 8 dpf; and we were unable to generate a homozygous *spout1/cenp-32* null strain in *C. elegans* (Supplementary Fig. 18). Furthermore, mice homozygous for knockout of the overlapping *Spout1/Cenp-32* and *EndoG* loci exhibit embryonic lethality before implantation (MGI: 106544)[35]. Therefore, all evidence indicates that *SPOUT1/CENP-32* is an essential gene for organismal viability[17].

**Fig. 8 | SPOUT1/CENP-32 patient variants lead to alterations of spindle organisation. A** Representative fluorescence images of a Control (siRNA C) or SPOUT1/CENP-32 (siRNA C32) siRNA rescue assay in stable U2OS cell lines with inducible expression of different SPOUT1/CENP-32-GFP constructs. Conditions with doxycycline only; conditions without doxycycline in Fig. S17A. **B, C** Quantification for the analysis of the centrosome detachment phenotype (% of cells with detached centrosomes; **B**) and chromosome segregation errors (% of micronuclei; **C**) in inducible U2OS cell lines expressing SPOUT1/CENP-32 WT-GFP or the SPOUT1/CENP-32 patient variants. For easy direct comparison with the SPOUT1/CENP-32 WT conditions shown in Fig. 6D are also shown in this figure. Data are representative of a minimum of three biological replicates, $n \geq 70$ cells analyzed in total per treatment, mean ± SD, n = 3. Scale bar, 10 μm. Two-sided Fisher's exact test with Bonferroni correction (*, $P \leq 0.05$; **, $P \leq 0.01$; ***, $P \leq 0.001$; ****, $P \leq 0.0001$). Exact p-values for the detachment centrosome phenotype (B) vs WT condition are: N86D siRNA C +Dox=3.1E-03, N86D siRNA C32 +Dox=4.3E-30, G98S siRNA C +Dox=1.9E-02, G98S siRNA C32 +Dox=4.9E-33, G244S siRNA C32 +Dox=4.8E-15, T289M siRNA C32 +Dox=1.8E-25, G293S siRNA C32 +Dox=2.7E-30, T353M siRNA C +Dox=3.3E-07, T353M siRNA C32 +Dox=4.1E-28. Exact p-values for the micronuclei phenotype (C) vs WT condition are: N86D siRNA C +Dox=1.3E-02, N86D siRNA C32 +Dox=2.9E-04, G98S siRNA C32 -Dox=1.9E-07, G98S siRNA C +Dox=2.5E-04, G98S siRNA C32 +Dox=2.1E-05, G244S siRNA C32 +Dox=1.6E-15, T289M siRNA C32 +Dox=1.2E-24, G293S siRNA C32 +Dox=1.3E-03, T353M siRNA C +Dox=1.1E-04, T353M siRNA C32 +Dox=5.5E-37. Data points for each of the replicates are shown. Source data are provided as a Source Data file. **D** Cell viability evaluation of inducible U2OS cell lines expressing SPOUT1/CENP-32 WT-GFP or SPOUT1/CENP-32 patient variants assessed by the MTT assay. For easy direct comparison with the SPOUT1/CENP-32 variant conditions, the SPOUT1/CENP-32 WT conditions shown in Fig. 6E are also shown in this figure. Three independent biological replicates, mean ± SEM. WT siRNA C -Dox: n = 26; WT siRNA C32 -Dox; n = 26; WT siRNA C +Dox: n = 25; WT siRNA C32 +Dox: n = 27; N86D siRNA C -Dox: n = 22; N86D siRNA C32 -Dox: n = 25; N86D siRNA C +Dox: n = 27; N86D siRNA C32 +Dox: n = 22; G98S siRNA C -Dox: n = 27; G98S siRNA C32 -Dox: n = 26; G98S siRNA C +Dox: n = 27; G98S siRNA C32 +Dox: n = 24; G244S siRNA C -Dox: n = 25; G244S siRNA C32 -Dox; n = 27; G244S siRNA C +Dox: n = 27; G244S siRNA C32 +Dox: n = 27; T289M siRNA C -Dox: n = 18; T289M siRNA C32 -Dox: n = 18; T289M siRNA C +Dox: n = 18; T289M siRNA C32 +Dox: n = 18; T353M siRNA C -Dox: n = 18; T353M siRNA C32 -Dox: n = 18; T353M siRNA C +Dox: n = 18; T353M siRNA C32 +Dox: n = 18; G293S siRNA C -Dox: n = 27; G293S siRNA C32 -Dox: n = 27; G293S siRNA C +Dox: n = 27; G293S siRNA C32 +Dox: n = 27. Non-parametric ANOVA with Kruskal-Wallis followed by Dunn's multiple comparisons test (*, $p \leq 0.05$; **, $p \leq 0.01$; ***, $p \leq 0.001$; ****, $p \leq 0.0001$). Exact p-values vs WT siRNA C -Dox condition are: WT siRNA C32 -Dox=1.4E-04, N86D siRNA C32 -Dox=2.4E-06, N86D siRNA C32 +Dox=4.6E-06, G98S siRNA C32 -Dox=8.7E-05, G98S siRNA C32 +Dox=2E-07, G244S siRNA C32 -Dox=1E-07, G244S siRNA C +Dox: 6.8E-03, G244S siRNA C32 +Dox=1E-07, T289M siRNA C32 -Dox=1E-07, T289M siRNA C32 +Dox: 1E-07, G293S siRNA C32 -Dox=1E-07, G293S siRNA C32 +Dox=1E-07, T353M siRNA C32 -Dox= 1E-07, T353M siRNA C +Dox=1.4E-02, T353M siRNA C32 +Dox=1E-07. Source data are provided as a Source Data file.

---

Our structure/function analysis reveals that SPOUT1/CENP-32 possesses a SPOUT methyltransferase domain. In vitro assays confirm that SPOUT1/CENP-32 is indeed an active RNA-methyltransferase. Importantly, this RNA methyltransferase activity of SPOUT1/CENP-32 is crucial for mitotic spindle assembly and accurate chromosome segregation. Interestingly, most *SPOUT1/CENP-32* bi-allelic variants found in patients are located close to the catalytic site, perturbing the methyltransferase activity of SPOUT1/CENP-32 in vitro and result in the detachment of centrosomes from mitotic spindle poles in a human cell line. Of note, the recurrent variants N86D, G98S and T353M showed the most significant defects in the functional studies, which correlate with in silico prediction scores (Supplementary Data 2). The G98S variant significantly reduced methyltransferase activity in vitro and failed to rescue the morpholino-induced head size phenotype in zebrafish, and caused chromosome segregation defects in cultured cells. This variant was found in 9/21 families in the cohort. Given that the G98S variant is located at the dimeric interface of the protein, it further highlights and supports the importance of dimerization for the enzymatic activity of the protein.

The strongest cellular phenotypes were observed for the T353M variant, seen in 3/21 families in the cohort, which showed an increased number of cells with detached centrosomes and an increased number of micronuclei. In addition to these cellular phenotypes, the reduction in head size phenotype in zebrafish was only partially rescued by the T353M variant, suggesting that it is a hypomorph, with some residual activity. Thus, both assays were consistent with the pathogenicity of the T353M variant.

Tethering of chromosomes to spindle poles is crucial to maintain the bipolar architecture of the mitotic spindle required for accurate chromosome segregation. Several molecular players, including dynein-dynactin, NuMA, CDK5RAP2/centrosomin, WDR62, and HSET have been implicated in centrosome-spindle pole attachment[50–59]. Depletion of WDR62 or CDK5RAP2 disrupts centrosome attachment to spindle poles but does not affect centrosomal nucleation of microtubules[50,51]. These observations suggest that centrosomal microtubules alone are insufficient to connect centrosomes to spindle poles, and the centrosome-spindle pole connection is likely maintained through physical tethering. Interestingly, SPOUT1/CENP-32 depletion did not abolish the centrosome association of CDK5RAP2, a component suggested to physically tether centrosome to spindle poles, but changed the abundance of CG-NAP, a PCM component, which failed to accumulate on the dissociated centrosomes in SPOUT1/CENP-32 depleted cells[16]. However, CG-NAP depletion alone did not phenocopy SPOUT1/CENP-32 depletion, suggesting that SPOUT1/CENP-32 regulates multiple centrosome components essential for centrosome-spindle pole attachment through mechanisms that are yet to be determined.

The precise mechanism by which SPOUT1/CENP-32 RNA methyltransferase activity ensures centrosome-spindle pole tethering remains to be elucidated. We hypothesize that SPOUT1/CENP-32 catalytic activity modulates the abundance and interactions of centrosomal proteins or RNAs essential for tethering centrosomes to spindle poles. RNA methyltransferases are known to regulate RNA stability, localization, and translation efficiency[60]. Most of the well characterized SPOUT MTases are involved in 2'-O-ribose methylation, which is mostly found in ribosomal and small nuclear RNAs, that mainly occurs in functionally important regions of the ribosome and spliceosome[41]. Moreover, RNA and RNA-binding proteins have also been implicated in centrosome function, as well as in the formation of the mitotic spindle[61–70]. Some examples include NSUN2, a nucleolar RNA MTase required for spindle formation[63] and RAE1, an RNA-binding protein that regulates spindle assembly in an RNA-dependent manner[69]. In support of our hypothesis, the SPOUT1/CENP-32 homolog in fission yeast was shown to interact with a newly identified centrosomal protein, Wdr8, which is involved in anchoring microtubules to the centrosome[71,72] and with dynactin 4 (DCNT4), a subunit of the dynactin complex[73]. Future studies will aim to elucidate the mechanistic links between SPOUT1/CENP-32 methyltransferase activity and centrosome-spindle pole tethering.

Besides the centrosome detachment phenotype, SPOUT1/CENP-32 depleted cells also showed chromosome segregation errors, as indicated by the formation of micronuclei. Accurate chromosome segregation is essential for maintaining genomic stability during mitosis, and aneuploidy resulting from chromosome mis-segregation is a cancer hallmark considered to be a driving force for tumorigenesis and tumor evolution[74,75]. However, we karyotyped >50 metaphase cells in peripheral blood cultures from individual A-III-2, but no aneuploidy was detected (Supplementary Fig. 19). It is possible that those aneuploid cells are eliminated by apoptosis in vivo in humans. Further studies will be required to determine whether individuals with *SPOUT1/CENP-32*-associated neurodevelopmental disorder have an increased risk of developing aneuploidy in vivo or even cancers.

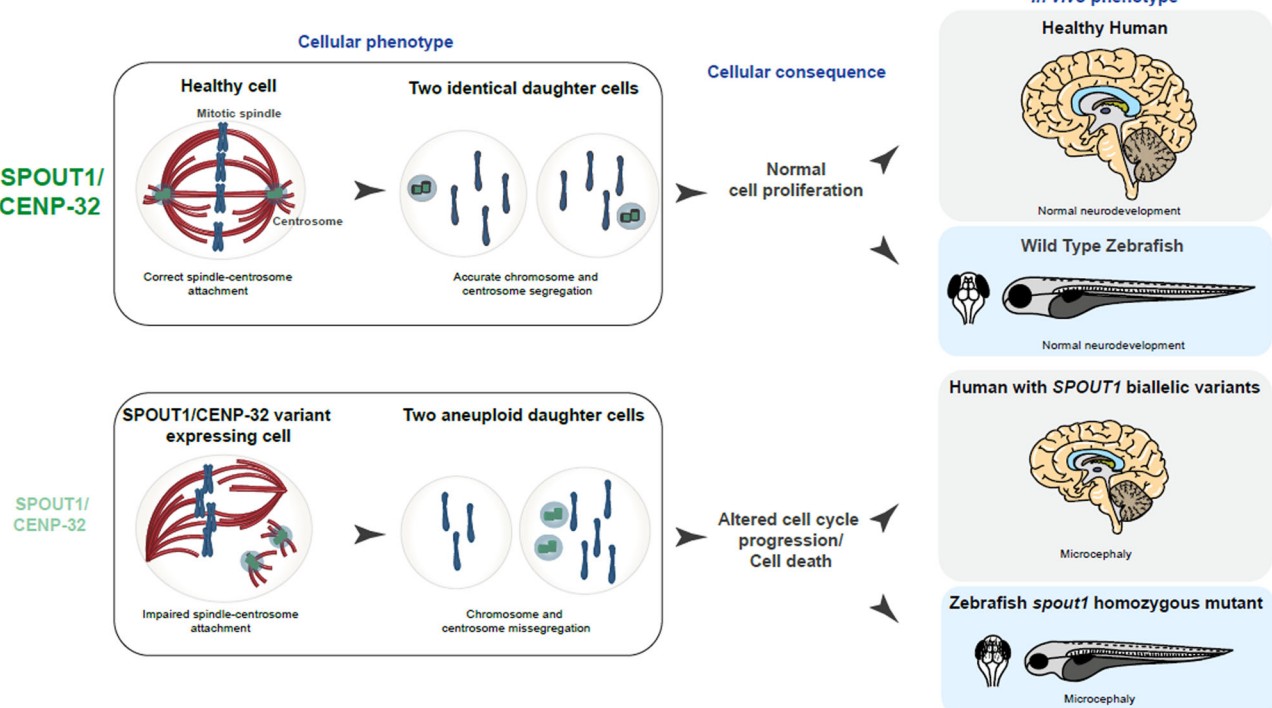

**Fig. 9 | *SPOUT1/CENP-32* bi-allelic variants disrupt centrosome-spindle pole tethering, causing chromosome segregation errors, altering cell cycle progress, and increasing apoptosis in neuronal progenitor cells.** Schematic summarizing the proposed differences between healthy cells and *SPOUT1/CENP-32* variant cells leading to aneuploid daughter cells, altered cell cycle/ cell death, and microcephaly in affected individuals and mutant zebrafish.

In conclusion, we report here that bi-allelic variants of the *SPOUT1/CENP-32* gene in humans are associated with an autosomal-recessive complex neurodevelopmental disorder: SpADMiSS. Through extensive experimental studies from biochemical and cellular functions to animal models, we demonstrate that the variants identified in affected individuals reduce the RNA methyltransferase activity of the SPOUT1/CENP-32 protein, causing centrosome detachment from spindle pole and chromosome mis-segregation. We propose that these defects delay the cell cycle and trigger apoptosis in neural progenitor cells, leading to neurodevelopmental defects in humans (Fig. 9). We recognize that factors including differences in SPOUT1/spout1 allele strength, genetic background, trans and cis factors likely contribute to differences in penetrance and expressivity observed for biallelic and heterozygous humans versus other species. Furthermore, one limitation of our study is the transient nature of our variant testing strategies in zebrafish and cells; generation and in-depth characterization of stable knock-in models will aid future mechanistic investigation.

## Methods

### Inclusion and ethics statement, recruitment of families, and evaluation

Written informed consent was obtained from all families. Clinical evaluation and genetic analyses were approved by respective institutional ethics committees from the following participating centers: Columbia University Medical Center (IRB# AAAR1159), Children's Hospital of Philadelphia (IRB #16-013231), Baylor College of Medicine (IRB# H-29697), and Nationwide Children's Hospital (IRB# 11-00215). All participating clinical centers were connected through GeneMatcher[18]. Sex of patients was determined based on self-reporting by the families and not considered in the study design. The patient cohort included both sexes, with 57% (16/28) females and 43% (12/28) males. Peripheral blood was obtained from participants and genomic DNA was extracted as per standard protocols. A pediatric epileptologist (T.T.S.) reviewed the genetic test results and clinical reports, and evaluated the magnetic resonance imaging (MRI) and electroencephalographic (EEG) recordings, where available.

### Genetic analyses

Exome sequencing (ES)/ Genome sequencing (GS) and variant calling were performed on DNA extracted from peripheral blood mononuclear cells using commercial exome libraries and Illumina, and standard clinical and research bioinformatic pipelines (see Supplementary case reports). Single nucleotide variants (SNVs) and indels were classified in accordance with the ACMG/AMP sequence interpretation guidelines[76]. Sanger dideoxy DNA sequencing was used to verify candidate variants identified by ES/GS, and determine inheritance and segregation within families. Additional genetic studies included karyotype analysis, chromosomal microarray, homozygosity mapping, and mitochondrial testing following standard protocols.

### Zebrafish husbandry and embryo maintenance

We performed all zebrafish experiments according to protocols approved by the Northwestern University Institutional Animal Care and Use Committee (IACUC protocol IS00016405). Adult zebrafish were housed in standard husbandry conditions in a 14 h light-10h dark cycle. We obtained fertilized embryos by natural mating of wild type (WT) NIH or *-1.4col1a1:egfp* transgenic reporter adults. The embryos were maintained in fresh E3 medium (0.581 g/L NaCl, 0.026 g/L KCL, 0.096 g/L CaCl$_2$.2H$_2$O, 0.163 g/L MgCl$_2$.2H$_2$O, and 0.0002% methylene blue) at 28°C until phenotype endpoint. Larvae were evaluated for apoptosis and cell cycle progression at 2 dpf and head size and brain structures at 3 dpf.

### CRISPR/Cas9 mediated genome editing in zebrafish embryos

We performed reciprocal BLAST of human SPOUT1/CENP-32 (NP_057474.2) against the translated zebrafish genome and identified a single *spout1/cenp-32* zebrafish ortholog (Ensembl ID: GRCz11: ENSRDARG00000019707) with 76% identity and 88% similarity. To

identify single guide (sg) RNA sites targeting *spout1/cenp-32*, we used CHOPCHOP[77–79] (Supplementary Table S1). Using synthetic oligos as template, we synthesized sgRNAs targeting exon 5 and exon 7 of ENSDART00000099535.5 using the GeneArt Precision gRNA Synthesis Kit (ThermoFisher Scientific), according to the manufacturer's guidelines. We injected 1 nL cocktail containing 100 pg sgRNA with or without 200 pg Cas9 protein (PNA Bio, CP01) into the cell of single-cell stage embryos. To determine sgRNA targeting efficiency, we extracted genomic DNA from individual F0 embryos at 2 dpf by proteinase K digestion (Life Technologies, AM2548). The region flanking the sgRNA target was PCR-amplified, denatured, slowly reannealed, and then migrated on a 20% polyacrylamide gel (ThermoFisher Scientific) to separate heteroduplexes[80] ($n = 2$ uninjected controls and $n = 6$ F0 mutants tested). To determine F0 mosaicism, we cloned the PCR-amplified product into TOPO-TA cloning vector (ThermoFisher Scientific), and individual colonies were sequenced ($n = 16$ colonies/embryo) with BigDye Terminator 3.1 chemistry (Applied Biosystems). To generate *spout1/cenp-32* mutants, we outcrossed F0 mosaic mutants with WT (NIH) adults and isolated F1 heterozygous animals carrying a 1 bp insertion (Ensembl ID: ENSDART00000099535.5 c.543_544insA; p.R215Kfs*7) in exon 7.

### DNA extraction from Dent's or paraformaldehyde fixed embryos

If required, fixed larvae were genotyped after wholemount imaging as described[81,82]. Briefly, larvae were placed in PCR tubes and incubated in 40 μl of lysis buffer (25 mM NaOH, 0.2 mM EDTA) for 45 min at 95°C. The tubes were cooled to 4°C before adding an equal volume of Hot-SHOT neutralization buffer (40 mM Tris-HCl); 5 μl DNA lysate was used for PCR.

### Quantitative real-time PCR (qPCR)

For qPCR analysis, we incrossed F1 heterozygous mutants and then decapitated larvae at 3 dpf. Tails were used for DNA extraction and genotyping; matched pooled heads ($n = 18$) were used for total RNA extraction using Trizol (ThermoFisher Scientific). We reverse transcribed 1 μg RNA to cDNA using QuantiTect reverse transcription kit (Qiagen) according to the manufacturer's protocol. qPCR was performed on a QuantStudio 3 real time PCR instrument (Applied Biosystems), using SYBR green chemistry (Applied Biosystems) according to manufacturer's instructions. The primer pairs used for qPCR annealed to exons 3-4 (upstream) and exon 11-12 (downstream) of the mutation site (listed in Supplementary Table S1). The thermocycler conditions were: 95 °C for 10 mins; 39 cycles (95°C for 15 sec and 60°C for 1 min); 95°C for 15 sec; 60°C for 15 sec; 95°C for 15 sec. The relative gene expression was calculated by 2-ΔΔCt and normalized to *b-actin*. Two biological and 6 technical replicates were performed for each sample.

### Transient suppression of *spout1/cenp-32* by Splice Blocking Morpholino (sb-MO)

We designed a morpholino (MO) antisense oligonucleotide targeting the *spout1/cenp-32* exon 4-intron 4 junction (e4i4: 5′-GTCAAAA-TAGCAGCTCACTTTGCGT-3′) to block the exon 4 splice donor site and standard control MO (5′-CCTCTTACCTCAGTTACAATTTATA-3′) (Gene Tools, LLC). We obtained fertilized embryos from natural matings of adult fish and injected 1 nL MO into the yolk at the one-to-four cell stage with increasing doses of e4i4 (3 ng, 6 ng, and 9 ng). To assess MO efficiency, embryos were harvested for total RNA extraction using Trizol reagent (ThermoFisher Scientific) at 2 dpf (20 embryos/condition, pooled) according to manufacturer's instructions. cDNA was generated with QuantiTect Reverse Transcription Kit (Qiagen) according to the manufacturer's protocol, followed by PCR with primers flanking the MO target locus (Supplementary Table S1), and migration of the amplified product on a 1% agarose gel. Resulting PCR bands were gel purified (QIAquick®

Gel Extraction Kit, Qiagen) and cloned into TOPO-TA cloning vector (ThermoFisher Scientific) and resulting colonies were Sanger sequenced the ($n = 4$/PCR band).

### Molecular cloning of human ORF *(SPOUT1/CENP-32)*, site directed mutagenesis and in-vitro transcription

We acquired a Gateway-compatible human *SPOUT1/CENP-32* full length open reading frame (ORF) entry clone (GenBank: BC033677.1 ThermoFisher Scientific; IOH40219) and cloned it into a Gateway-compatible pCS2+ destination vector via LR clonase-mediated recombination (ThermoFisher Scientific). Next, we used it as a template to generate mutant constructs harboring *SPOUT1/CENP-32* variants identified in affected individuals (p.G98S, p.T353M, p.N86D, p.G293S, p.R200W, p.R352C, p.G244S, p.L314F, p.Y339C, p.T289M and p.S249del) or in gnomAD (dbSNP ID: rs6478854, p.T130R; negative control) by site-directed mutagenesis as described[37] (Supplementary Table S1). The integrity of all constructs was confirmed by Sanger sequencing. To generate capped mRNA for rescue experiments, we linearized pCS2+ constructs with NotI and used the resulting template for in vitro transcription with the mMessage mMachine SP6 Transcription kit (ThermoFisher Scientific) according to manufacturer's guidelines. We used 100 pg *SPOUT1/CENP-32* mRNA with 6 ng MO for in vivo complementation studies.

### Zebrafish phenotyping

**Live imaging of zebrafish larvae and bright field morphometrics.** To assess the head size and body length of *spout1/cenp-32* models, we anesthetized larvae with 0.2 mg/mL Tricaine (MS-222) at 3 dpf prior to loading into a 50 ml conical tube attached to the Vertebrate Automated Screening Technology (VAST) Bioimager (Union Biometrica). We passed larvae sequentially through a 600 μm capillary in the stage-mounted detection platform on an AxioImager.M2m microscope (Zeiss). We created dorsal and lateral image templates, and imaged them live with VAST software (version 1.2.6.7) with >60% minimum similarity for the pattern-recognition algorithm. Bright field (Dorsal and lateral) images were acquired with the on-board camera and VAST software using default parameters as described[22,83,84]. Head size area and body length were measured on lateral bright field images with ImageJ (NIH) software (version 1.53a). Specifically, head size was arbitrarily defined with three consistent landmarks and connected with a continuous line from the posterior otolith (line 1) to the nearest dorsal point from the otolith (line 2) to the most anterior point of the head at the mouth (line 3), and returning to the posterior otolith. Body length was measured by drawing a straight line from the most anterior part to the most posterior part.

**Whole-mount TUNEL assay.** Terminal deoxynucleotidyl transferase biotin-dUTP nick labeling (TUNEL, ApopTag Red In Situ Apoptosis Detection Kit, Millipore Sigma) was used to detect apoptotic cells in whole-mount zebrafish larvae[24,29,85]. Briefly, embryos were dechorionated at 2 dpf and fixed in 4% paraformaldehyde (PFA) overnight at 4°C, then dehydrated in 100% methanol at -20 °C overnight. The larvae were then gradually rehydrated in methanol in PBS and 0.1% Tween (PBST) in the following percent volume/volume ratios: 75/25; 50/50; and 25/75 for 10 min each at room temperature (RT). After rehydration in methanol/PBS, the larvae were bleached for 10 min in bleaching solution (0.5% KOH and 3% $H_2O_2$ in PBST) before permeabilization with proteinase K (10 μg/ mL) and post-fixation with 4% PFA for 20 min at RT. The larvae were then treated with equilibration buffer (Millipore Sigma) for 1 h at room temperature followed by overnight incubation with TdT enzyme in a humidified incubator. The next day larvae were washed with PBST, and then treated with the anti-digoxigenin-rhodamine (Millipore Sigma) for 2 h at RT and washed three times for 10 min each with PBST and processed for imaging.

**Immunostaining on whole-mount zebrafish larvae.** Whole mount anti-phospho-histone H3 and acetylated tubulin immunostaining were performed as described[23,25,86]. In brief, embryos were dechorionated at 2 dpf (pHH3), or larvae at 3 dpf (acetylated tubulin), and were fixed overnight in 4% PFA or Dent's solution (80% MeOH + 20% DMSO), respectively. Larvae were then rehydrated in PBS with stepwise reduced concentration of methanol at RT as mentioned above, followed by bleaching for 10-15 mins described elsewhere[22,29]. The larvae were permeabilized with proteinase K (10 μg/ mL) before post fixation with 4% PFA for 20 mins at RT, and washed twice in IF buffer (1% BSA, 0.1% Tween-20 in 1xPBS) for 10 mins. The larvae were then incubated in blocking solution (IF buffer and 10% FBS) for 1 h, followed by overnight incubation in primary antibody (anti-pHH3, 1:500, Santa Cruz Biotechnology, sc-374669; anti-α-acetylated tubulin, 1:1,000 Sigma Aldrich, T7451) at 4 °C. After two washes in IF buffer, the larvae were then treated with secondary antibody (Alexa Flour 488 goat anti-rabbit IgG, 1:500; Alexa Flour 594 goat anti-mouse IgG, 1:500) in blocking solution for 2 hrs at RT. The embryos were then washed three times with IF buffer, and processed for imaging.

***spout1/cenp-32* mutant activity tracking.** We obtained progeny from heterozygous mutant adult in-crosses and distributed larvae at 3 dpf into a 96 well plate maintained at a temperature of 28°C. We recorded larval activity in the DanioVision platform (Noldus B.V.). The acclimatization time was 15 minutes, and the activity was recorded for the next 30 minutes. Larvae were genotyped following data acquisition and matched to activity values recorded for each well. Data were initially analyzed with EthoVision XT 14 (Noldus B.V.) and later normalized in Prism 10.

**Fluorescence microscopy and image processing.** We captured Z-stacked images of the fluorescent signal with a Nikon AZ100 microscope equipped with Nikon camera and Nikon NIS Elements software (TUNEL, pHH3, and α-acetylated tubulin). We counted apoptotic and proliferative cells in a consistently defined head region, using the image-based tool for counting nuclei (ICTN) plugin with the following parameters: width: 12, min distance: 10, and threshold: 1 in ImageJ (version 1.53a)[29]. We also measured the area of the optic tecta, and number of intertectal neurons crossing the midline between optic tecta as described[23,25].

**Mouse care and experimentation.** IHC experiments were performed on C3H/HeN (from Jackson Lab, USA) 8-10 week old pregnant mice. ISH experiments were performed on C57Bl/6 (from Charles River Laboratories, USA) 8-10 week old pregnant mice. All animal procedures adhered to protocols approved by the Bioethics Committee of Lobachevsky State University of Nizhny Novgorod (protocol No. 14, dated 19 January 2018). Mice were housed in a controlled environment with a 12-hour light and dark cycle, maintained at 18-23 °C and 40-60% humidity, with free access to pelleted food and water. The subjects used in the experiments were at embryonic stages E12.5 to E18.5, with stage E0.5 marking the day the vaginal plug was observed.

**In utero electroporation.** In utero electroporation was performed as previously described (https://doi.org/10.1093/cercor/bhab172) using a FemtoJet 4i microinjector (Eppendorf, Germany) and a NEPA21 electroporator (NEPA GENE, Japan). Following surgery, animals were kept under observation until full recovery and the embryos collected at the pregnancy stages indicated in the experiments. The plasmids used in this study are presented in section cloning of target vectors (Supplementary methods).

**In situ hybridization.** Sectioning, in situ hybridization, washing were performed as described (https://doi.org/10.1093/nar/gkad703). Three independent in situ analyses were performed for each stage. The

primers used to generate the RNA probes had the sequences as listed: Spout1 fw-GACCTCGAGAAGGGTTGAATGGCGAAAATGG, Spout1 rw-AATTAACCCTCACTAAAGGGCGGCCGCCACAGTTGACCAAAGAGCCA TG.

**Immunohistochemistry.** Immunohistochemistry and tissue processing were performed as previously described (https://doi.org/10.1016/j.ydbio.2006.06.040) The primary antibodies used were a-BrDU (mouse), a-GFP (goat), a-Ki67(rabbit).

**Image acquisition and processing.** The slides resulting from in situ hybridization were imaged on a Zeiss Axio Vert.A1. For the immunofluorescently stained tissue preparations, we used a LSM 800 (Carl Zeiss) confocal laser scanning system. For each electroporated litter, brain sections were matched for anteroposterior and lateromedial position of the electroporation site. Counting was performed blinded.

**Protein expression and purification of SPOUT1/CENP-32 for crystallization.** The methyltransferase domain of SPOUT1/CENP-32 (SPOUT1/CENP-32 71-376) was cloned into a pEC-K-3C-His vector as an N-terminally His-tagged protein with a 3 C site. The catalytic site variant (A356N) was obtained using the Quikchange site-directed mutagenesis procedure (Agilent). Both selenomethionine-labeled and native SPOUT1/CENP-32 71-376 were expressed in BL21 gold *E.Coli* strain by overnight induction at 20 °C or 18 °C with 0.35 mM IPTG, respectively. Cells were lysed by sonication on lysis buffer containing 20 mM Tris-HCl pH 8, 500 mM NaCl, 35 mM Imidazole and 2 mM beta-mercaptoethanol. The pre-cleared lysate was loaded onto a 5 ml HisTrap™ HP column (Cytiva), washed with 50 cv of lysis buffer, 20 cv of high salt buffer (20 mM Tris-HCl pH 8, 1 M NaCl, 35 mM Imidazole, 50 mM KCl, 10 mM $MgCl_2$, 2 mM ATP and 2 mM beta-mercaptoethanol) and eluted with elution buffer (20 mM Tris-HCl pH 8, 200 mM NaCl, 400 mM Imidazole and 2 mM beta-mercaptoethanol). Proteins were then cleaved overnight at 4 °C with 3 C protease while dialyzing against 20 mM Tris-HCl pH 8, 200 mM NaCl and 5 mM DTT. Proteins were further purified by size-exclusion chromatography in 20 mM Tris-HCl pH 8, 200 mM NaCl and 5 mM DTT (Superdex 75 10/300 GL, Cytiva).

**SPOUT1/CENP-32 crystallization, data collection, crystal structure solution and refinement.** Gel filtrated selenomethionine SPOUT1/CENP-32 71-376 and native SPOUT1/CENP-32 71-376 A356N were concentrated using Vivaspin™ Turbo 10 kDa MWCO concentrators to 16 – 20 mg/ml. Crystallization trials were performed with a nanoliter Crystal Gryphon robot (Art Robins) and grown by vapor diffusion using the Morpheus™ screen (Molecular Dimensions). SPOUT1/CENP-32 71-376 proteins crystallized in several conditions of the Morpheus screen. The best resolution for the selenomethionine-labeled SPOUT1/CENP-32 71-376 was obtained from the crystals grown in conditions containing: 0.06 M $MgCl_2$, $CaCl_2$ or 0.09 M NaF, NaBr, NaI with 0.1 M Imidazole, MES (acid) pH 6.5 and 30 % Ethylene glycol, PEG 8 K. To obtain the SPOUT1/CENP-32 71-376 – SAM structure, selenomethionine-labeled SPOUT1/CENP-32 71-376 crystals were soaked with 10 time's molar excess of SAM (ab142221, Abcam). The crystals that diffracted better for native SPOUT1/CENP-32 71-376 A356N protein grew in 0.09 M NaF, NaBr, NaI; 0.1 M Sodium HEPES, MOPS (acid) pH 7.5 and 30 % Ethylene glycol, PEG 8 K.

Diffraction data were collected on beamlines i03 (SPOUT1/CENP-32 71-376 SAH and SPOUT1/CENP-32 71-376 SAM) and i04-1 (SPOUT1/CENP-32 71-376 A356N) at the Diamond Light Source (Didcot, UK). All datasets were processed using the software pipeline available at Diamond Light Source that utilizes XDS, CCP4, CCTBX, AutoPROC and STARANISO[87–92]. SPOUT1/CENP-32 71-376 SAH structure was determined using Single Anomalous Dispersion method using Se-Met phasing. SPOUT1/CENP-32 71-376 SAM and SPOUT1/CENP-32 71-376

A356N structures were determined by molecular replacement with the program PHASER[93] using the coordinates of SPOUT1/CENP-32 71-376 SAH structure. Structures were refined using the PHENIX and CCP4 suite of programs[94,95]. Model building was done using COOT[96] and figures were prepared using PyMOL (http://www.pymol.org). Data collection, phasing and refinement statistics are shown in Supplementary Table S3.

The structural coordinates and structure factors reported in this paper have been deposited in the PDB (http://www.rcsb.org/) with the following accession numbers: 8QSU for SAH-bound CENP-32 71-376, 8QSV for SAM-bound CENP-32 71-376 and 8QSW for CENP-32 71-376 catalytic mutant (A356N).

**Methyltransferase assays.** NSUN6 was cloned in a pET-His6-MBP vector (pET His6 MBP TEV cloning vector with BioBrick polycistronic restriction sites (9 C) was a gift from Scott Gradia; Addgene plasmid #48286; http://n2t.net/addgene:48286; RRID:Addgene_48286) and purified as previously described[45] with a few variations. Briefly, His6-MBP-NSUN6 was expressed in *E. coli* BL21 (DE3) pLysS cells by induction with 0.5 mM IPTG at 18 °C for 16 hrs. Cells were lysed by sonication in a lysis buffer containing 30 mM KPi pH 7, 300 mM KCl, 10 % Glycerol, 25 mM Imidazole and 1 mM beta-mercaptoethanol. The lysate was then cleared by centrifugation at 61,240 x *g* for 50 min 4 °C and loaded onto a 5 ml HisTrap™ HP column (Cytiva). The protein-bound column was washed with 60 cv of lysis buffer and 40 cv of high salt buffer (30 mM KPi pH 7, 500 mM KCl, 10 % Glycerol, 35 mM Imidazole and 1 mM beta-mercaptoethanol). Protein was eluted with elution buffer containing 30 mM KPi pH 7, 300 mM KCl, 10 % Glycerol, 200 mM Imidazole and 1 mM beta-mercaptoethanol and the pool of elutions was dialysed overnight at 4 °C into storage buffer (30 mM KPi pH 7, 100 mM KCl, 50 % Glycerol, 1 mM DTT and 0.1 mM EDTA).

Survivin was used as a negative control in our methyltransferase assays as it does not have methyltransferase activity. Full length Survivin was cloned into a pEC-K-3C-His vector as an N-terminally His-tagged protein with a 3 C cleavage site. The vector was transformed in *E. coli* BL21 Gold strain and grown in Super broth media at 37°C until O.D 1-1.5. Cultures were induced over night at 18 °C with 0.35 mM IPTG. Cells were lysed in a buffer containing 20 mM Tris-HCl pH 8, 150 mM NaCl, 25 mM imidazole and 2 mM 2-mercaptoethanol. The protein was purified by affinity chromatography using a 5 ml HisTrap™ HP column (Cytiva). The protein-bound column was washed with 40 column volumes of lysis buffer followed by 20 column volumes of high salt buffer (20 mM Tris-HCl pH 8, 1 M NaCl, 25 mM imidazole and 2 mM 2-mercaptoethanol). The protein was eluted and dialysed overnight at 4 °C in 20 mM Tris-HCl pH 8, 150 mM NaCl, 2 mM DTT while cleaving with 3 C protease. The cleaved protein was concentrated with a Vivaspin™ 10 kDa MWCO concentrator (Millipore), and the concentrated protein was loaded onto a Superdex 200 increase 10/300 GL column (Cytiva) equilibrated with 20 mM HEPES pH 7.5, 100 mM NaCl and 2 mM DTT.

Full length SPOUT1/CENP-32 WT, A356N, T130R, and patient variants (N86D, G98S, G293S, G244S, T289M, T353M) were cloned as described above in a pEC-K-His vector as N-terminally His-tagged proteins and expressed using the *E.Coli* BL21 Gold strain in 250 ml of Super Broth media. Harvested cells were lysed by sonication in lysis buffer containing 20 mM Tris-HCl pH 8, 500 mM NaCl, 35 mM Imidazole and 2 mM beta-mercaptoethanol, supplemented with RNase, DNase and PMSF. Lysates were incubated with 2 ml of HisPur™ Ni-NTA resin (Thermo Fisher Scientific) for 2 h at 4 °C and protein-bound beads were washed with 30 cv of lysis buffer followed by 20 cv of high salt buffer (20 mM Tris-HCl pH 8, 1 M NaCl, 35 mM Imidazole and 2 mM beta-mercaptoethanol). Proteins were eluted with 20 mM Tris-HCl pH 8, 300 mM NaCl, 400 mM Imidazole and 2 mM beta-mercaptoethanol and pools of elutions were dialysed over night against 20 mM HEPES pH 8, 300 mM NaCl, 1 mM DTT and 20 % Glycerol. Methyltransferase assays were performed straight after dialysis.

The total RNA used as a substrate in the methyltransferase assays was isolated from asynchronous SPOUT1/CENP-32 depleted (using predesigned siRNA oligos directed against human SPOUT1/CENP-32 (target sequence: TCGCAGGACCCTCGCACCAAA; Qiagen) or NSUN6 depleted (target sequence: TTCCTGTTATTGGACCCAGAA)[45], U2OS cultures via the TRIzol™ (Thermo Fisher Scientific) method that involves homogenisation of the cell lysate, phase separation, RNA precipitation and RNA washes. The RNA hairpin of GAPDH mRNA used as a substrate in the methyltransferase assays was synthesized by Integrated DNA Technologies: GCCCCCUCUGCUGAUGCCCCCAU GUUCGUCAUGGGGUGUGAA.

Methyltransferase assays were performed using the MTase-Glo™ Methyltransferase assay (Promega Corporation) following manufacturer's instructions. Briefly, assays were performed at room temperature with reaction buffer containing 20 mM Tris-HCl pH 8, 50 mM NaCl, 1 mM EDTA, 3 mM MgCl₂, 0.1 mg/ml BSA and 1 mM DTT in flat bottom non-binding 96 well microplates (Greiner BioOne). The proteins were incubated with the substrates (either total RNA or *GAPDH* mRNA oligo) for 30 min, when 0.5 % TFA was used to stop the reaction. 5 µL of 6× MTase-Glo Reagent were then added to each well and incubated for 30 min at room temperature. For the generation of the bioluminescent signal, 30 µL of MTase-Glo detection reagent were added to the reaction and incubated for a further 30 min at room temperature. Luminescence was measured using SpectraMax M5 Multi-mode microplate reader (Molecular Devices). Experiments were performed at least in triplicate, and values expressed as means ± the standard error of the mean (SEM). Data were analyzed using nonlinear regression to determine Michaelis-Menten kinetics in GraphPad Prism version 7.0.

**Size exclusion Chromatography.** Size exclusion chromatography (SEC) experiments with purified SPOUT1/CENP-32 full length recombinant proteins to test the oligomerisation state of the SPOUT1/CENP-32 variants were performed on an AKTA Pure 25 HPLC unit (Cytiva) with a Superdex 200 10/300 GL 24 ml column (Cytiva) at 4 °C in 20 mM Tris-HCl pH 7.5, 200 mM NaCl, 2 mM DTT. 0.5-ml fractions were collected with a 0.2–column volume delayed fractionation setting. UV 280- and 260-nm wavelengths were monitored.

**Mass photometry.** Microscope coverslips (no. 1.5, 24 × 50 mm) were cleaned with 100% isopropanol in a sonic bath and dried. Silicone gaskets (GBL103250-10EA; Grace BioLabs) were placed on the coverslips. Purified SPOUT1/CENP-32 full length recombinant proteins were diluted to 25 nM in buffer containing 20 mM HEPES pH 7.5, 500 mM NaCl, 1 mM DTT and 20 % glycerol immediately before mass photometry measurements. Diluted protein was measured following manufacturer's instructions using default settings. All data was acquired using a Two^MP mass photometer instrument (Refeyn) and AcquireMP software (Refeyn, v2023 R1.1). Data was analysed using Discover^MP software (Refeyn, v2023 R2).

**DNA constructs and generation of SPOUT1/CENP-32-GFP stable cell lines.** For CRISPR-mediated double-strand break site formation, 54-bp DNA fragments were generated by annealing two complementary single-stranded oligonucleotides. The fragments were inserted into the AgeI (#R3552, New England BioLabs [NEB]) site of pTORA14[97] using the In-Fusion HD Cloning Kit, yielding pTOR-A14AAVS1 which were used to induce DNA cleavage at adeno-associated virus integration site 1 (AAVS1). The DNA sequences targeted by the guide RNAs are listed in Supplementary Table S4.

pDEST243NGFP was generated by inserting a DNA fragment encoding EGFP amplified (primers are listed in Supplementary Table S5) from pDEST131NGFP[97] by PCR, between the BglII (#R0144,

NEB) and MluI (#R0198, NEB) sites of pMK243[98] with the In-Fusion HD Cloning Kit (#Z9648N, Takara Bio). The DNA fragment encoding human SPOUT1/CENP-32[71-end] were PCR-amplified (primers are listed in Supplementary Table S5) from pEGFP-N1-CENP32_71-end. The fragments were inserted into pDEST243NGFP between the BglII (#R0144, NEB) and SalI (#R0138, NEB) sites using the In-Fusion HD Cloning Kit, yielding pDEST243CENP32[71-end]. The resultant plasmids were used to generate a human cell line expressing the SPOUT1/CENP32[71-end]:GFP fusion protein via CRISPR-mediated homologous recombination (HR) at AAVS1. The fragments generated from the digestion of pEGFP-N1-CENP-32 WT, pEGFP-N1-CENP-32 A356N, pEGFP-N1-CENP-32 G244S, pEGFP-N1-CENP-32 T289M, pEGFP-N1-CENP-32 T353M or pEGFP-N1-CENP32 G293S by BglII and AgeI were inserted into pDEST243CENP32[71-376] between the BglII and AgeI sites using the T4 DNA ligase (#M0202, NEB), yielding pDEST243CENP32[WT], pDEST243CENP32[A356N], pDEST243CENP32[G244S], pDEST243CENP32[T289M], pDEST243CENP32[T353M], pDEST243CENP32[G293S]. The resultant plasmids were used to generate a human cell line expressing the CENP32[WT], CENP32[A356N], CENP32[G244S], CENP32[T289M], CENP32[T353M] or CENP32[G293S]:EGFP fusion protein via CRISPR-mediated homologous recombination (HR) at AAVS1.

A U2OS conditional overexpressing (cOX) SPOUT1/CENP-32 and its derivatives cell line, which is the parental cell line of *CENP-32*:cOX cells, was generated by transfecting U2OS cells with pTORA14AAVS1 and pDEST243CENP32 using Lipofectamine LTX Reagent (#A12621, Thermo Fisher Scientific) followed by puromycin selection (1 µg/mL, #10-2100, Focus Biomolecules). The genome editing was confirmed by PCR-amplification from the genomic DNA using Tks Gflex DNA Polymerase (#R060A, Takara Bio) and the primers listed in Supplementary Table S5. Also the expression of the GFP fusion protein was confirmed by microscopic observation.

### In vitro rescue experiments and Immunofluorescence microscopy.
The U2OS human osteosarcoma cells were routinely maintained in DMEM (Thermo Fisher Scientific) supplemented with 10% fetal bovine serum and penicillin/streptomycin (10,000 U/ml; Thermo Fisher Scientific).

Lipofectamine RNAimax was used for depletion of endogenous SPOUT1/CENP-32 using predesigned siRNA oligos directed against human *SPOUT1/CENP-32* (target sequence: TCGCAGGACCCTCGCAC-CAAA; Qiagen). Luciferase targeting was used as a control (5′-CGUACGCGGAAUACUUCGAdTdT-3′; All Star, Qiagen)[99]. Cells were plated in glass coverslips in 12 well plates and 16 h after plating they were transfected with 15 pmols of siRNA oligos and incubated for 48 hrs. For analysis of the centrosome detachment phenotype, cells were fixed with ice-cold methanol for 5 min rehydrated with PHEM buffer (60 mM PIPES pH 6.9, 25 mM HEPES, 10 mM EGTA and 2 mM MgCl₂) and blocked with 1 % BSA in PHEM buffer for 30 min at RT. Cells were stained with mouse anti-tubulin (B512; 1:1000; T5168; Sigma) and rabbit anti-pericentrin (1:1000; ab4448; Abcam). The secondary antibodies used were donkey anti-mouse Cy5 (1:300, 715-175-151; Jackson Laboratories) and goat anti-rabbit TRITC (1:300; 111-025-006; Jackson Laboratories). Slides were mounted with Vectashield with DAPI (H-1200, Vector Labs). A minimum of 70 cells per condition were quantified. The same slides were used to quantify the percentage of cells with micronuclei. Imaging was performed at room temperature using a wide-field DeltaVision Elite (Applied Precision) microscope with Photometrics Cool Snap HP camera and 100× NA 1.4 Plan Apochromat objective with oil immersion (refractive index = 1.514) using SoftWoRx 3.6 (Applied Precision) software. The acquired images were processed by constrained iterative deconvolution using SoftWoRx 3.6 software package (Applied Precision). Shown images are deconvolved and maximum-intensity projections. Statistical significance of the data was established by a two-sided Fisher's exact test and a Bonferroni correction using R studio.

### Cell viability assays.
U2OS human osteosarcoma cells were plated in 96 well plates and 16 h after plating they were transfected with 3.5 pmols of siRNA oligos (Control or SPOUT1/CENP-32) using Lipofectamine siRNA max. 72 h after siRNA transfection, 10 µl of MTT labeling reagent were added to each well (Cell Proliferation kit I-MTT, Roche) and incubated for 4 h at 37 °C and 5 % $CO_2$. Then 100 µl of Solubilization buffer (Cell Proliferation kit I-MTT, Roche) were added to each well and incubated at 37 °C for 1 h to allow for total solubilization of the purple formazan crystals. Microplates were evaluated using the Spectramax M5 microplate reader (Molecular devices) at 570 nm with a reference wavelength of 700 nm. Statistical significance of the data was established by a non-parametric ANOVA with Kruskal-Wallis followed by Dunn's multiple comparisons test using GraphPad Prism version 7.0.

### Size Exclusion chromatography coupled to multi-angle light scattering.
Size-exclusion chromatography (ÄKTA-Micro™, GE Healthcare) coupled to UV, static light scattering and refractive index detection (Viscotek SEC-MALS 20 and Viscotek RI Detector VE3580; Malvern Instruments) was used to determine the molecular mass of SPOUT1/CENP-32 71-end in solution. 200 µl of 1 mg/ml SPOUT1/CENP-32 71-end ($\partial A_{280\ nm}/\partial c = 0.667$ AU.ml/mg) was run on a Superdex 200 increase 10/300 GL size-exclusion column pre-equilibrated in 50 mM HEPES pH 8.0, 350 mM NaCl and 5 mM DTT at 22 °C with a flow rate of 0.5 ml/min. Light scattering, refractive index (RI) and $A_{280\ nm}$ were analysed by a homo-polymer model (OmniSEC software, v5.02; Malvern Instruments) using the parameters stated for SPOUT1/CENP-32 71-end, $\partial n/\partial c = 0.185$ ml/g and a buffer RI value of 1.335. The mean standard error in the mass accuracy determined for a range of protein–protein complexes spanning the mass range of 6–600 kDa is ± 1.9%.

### Western Blot.
To determine SPOUT1/CENP-32 levels after siRNA oligo treatment and to test the expression levels of each of the SPOUT1/CENP-32 inducible stable cell lines, U2OS cell lines were transfected in 12-well dishes as described above for the rescue experiments, lysed in 1× Laemmli buffer, boiled for 5 min, and analyzed by SDS-PAGE followed by Western blotting. The antibodies used for the immunoblot were rabbit anti-CENP-32 (HPA022990; 1:250; Atlas Antibodies), rabbit anti-GFP (ab290; 1:5,000, Abcam) and mouse anti-tubulin (ab18251; 1:10,000; Abcam). Secondary antibodies used were goat anti-mouse 680, donkey anti-rabbit 800, and donkey anti-mouse 800 (LI-COR) at 1:2,000. Immunoblots were imaged using the Odyssey CLx system, and band intensities were quantified using ImageJ, uncalibrated OD values. Uncalibrated OD values were corrected by the corresponding tubulin levels (loading control) and normalized to siRNA control values. Three experimental replicates were analyzed.

### Statistics & reproducibility
Statistical analyses for the assays with the U2OS cell lines were performed using GraphPad Prism version 7 (GraphPad Software). Data throughout the article are expressed as mean ± SD or mean ± SEM, as stated in the figure legends. Comparison between two nominal variables were analyzed by two-tailed Fisher's exact test with Bonferroni correction. Comparison between multiple groups were tested using analysis of variance, non-parametric ANOVA with Kruskal-Wallis followed by Dunn's multiple comparisons test. A minimum of three biological replicates were performed for all experiments. Data were considered statistically different at $p \leq 0.05$ with a single asterisk, at $p \leq 0.01$ with two asterisks, at $p \leq 0.001$ with three asterisks, and at $p \leq 0.0001$ with four asterisks.

Each zebrafish experiment was performed at least twice, and measurements were taken with the investigator blinded to experimental condition (see Supplementary Table S2 for experimental data). All zebrafish morphometric data were analyzed using Prism 9 software

(GraphPad, San Diego, California, USA). Data were first evaluated using descriptive statistics to check equality of variances and normal distributions (e.g., D'Agostino-Pearson omnibus test). When necessary, we used a non-parametric ANOVA (Kruskal-Wallis test) followed by Dunn's multiple comparisons test. Correction for multiple comparisons was accomplished using the original method of Benjamini and Hochberg (Benjamini and Hochberg, 1995) to control the False Discovery Rate (FDR) which was set at 5% (Q = 0.05).

Statistical analysis for mouse embryo experiments was performed using the Prism software (GraphPad), in accordance with the nature of the experimental setup and employing the tests indicated in the experiments.

### Reporting summary
Further information on research design is available in the Nature Portfolio Reporting Summary linked to this article.

## Data availability
The *SPOUT1/CENP-32* variants identified in this study have been deposited in the ClinVar database (https://www.ncbi.nlm.nih.gov/clinvar/) under the following variation IDs: 2274948 (c.598 C > T, p.Arg200Trp), 3236473 (c.730 G > A, p.Gly244Ser), 3236474 (c.877 G > A, p.Gly293Ser), 3236475 (c.292 G > A, Gly98Ser), 3236476 (c.1016 A > G, p.Tyr339Cys), 3236477 (c.1054 C > T, p.Arg352Cys), 3252018 (c.1058 C > T, p.Thr353Met), 3252019 (c.940 C > T, p.Leu314Phe), 547052 (c.102dup, p.Trp35Metfs), 3241959 (c.766 G > C, p.Ala256Pro), 3241960 (c.590 G > A, p.Arg197Gln), 3242264 (c.866 C > T, p.Thr289Met), 3242263 (c.744_746del, p.Ser249del), and 547051 (c.256 A > G, p.Asn86Asp). Individual sequencing data from ES or GS may be subject to restrictions due to patient confidentiality. Crystal structures of SPOUT1/CENP-32 with SAH, SAM and the catalytic site mutant (A356N) have been deposited in Protein Data Bank (PDB: http://www.rcsb.org/) with the following accession numbers: 8QSU, 8QSV and 8QSW, respectively. The representative fluorescence images (Figs. 6, 8, S15 and S17) generated in this study have been deposited in Figshare [https://figshare.com/projects/RNA_methyltransferase_SPOUT1_CENP-32_links_mitotic_spindle_organization_with_the_neurodevelopmental_disorder_SpADMiSS/218425]. Any additional information is available upon reasonable request to the corresponding authors. Source data are provided with this paper.

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

## Acknowledgements

We thank the families for consenting to participate in this study. We thank Elizabeth LeClair for assistance on statistical analyses of zebrafish data; and Jaewon Lee for help with illustrations in Fig. 9. This work was supported by a grant from the US National Institute of Health (NIH) (R01MH106826) to EED, in part by US NIH (NS105078) to JRL and US NIH (U01 HG011758) to JRL and JEP. SK is funded by an International Research Support Initiative Program (IRSIP) fellowship from the Higher Education Commission of Pakistan. EED is the Ann Marie and Francis Klocke, MD Research Scholar. IS and WCE are funded by Wellcome Principal Research Fellowship to WCE (grant no. 107022). We thank Paula Sotelo-Parrilla and Anjitha Gireesh for their help with the mass photometry experiments. We thank Natalia Kochanova and Shaun Webb for their help with the statistical analyses of the in vitro U2OS rescue experiments and we also thank Elizabeth Blackburn for helpful discussions on the Methyltransferase assays. We thank Juan C Celis Suescún, Anastasia Filat'eva, Polina Anisimova for assistance with mouse experiments. Mouse experiments were supported by the Ministry of Science and Higher Education of the Russian Federation (project no. FSWR-2023-0029). MAA is funded by a grant from the Medical Research Council (MRC, United Kingdom; MR/X001245/1) to AAJ. AAJ and his team are co-funded by the European Union (ERC Advanced Grant, CHROMSEG, 101054950). Views and opinions expressed are, however, those of the authors only and do not necessarily reflect those of the European Union or the European Research Council. Neither the European Union nor the granting authority can be held responsible for them. DM was supported by a medical genetics research fellowship program through the US NIH (T32 GM007526-42). JEP was supported by NHGRI K08 HG008986. TM was supported by the Uehara Memorial Foundation. DP is supported by Doris Duke Charitable Foundation with grant# 2023-0235, and NINDS NS125126-01A1. SNW was supported in part by K08NS119567. BPD was funded by Instituto de Salud Carlos III (grant numbers PI24/01083 and FORT23/00034).

## Author contributions

Conceptualization: A.V.D., V.T., S.N.W., A.A.J., E.E.D., J.L.; Funding acquisition: A.A.J., E.E.D., J.R.L., J.E.P., S.N.W., V.T., W.C.E., D.T.; Investigation: A.V.D., M.A.A., S.K., R.M., T.T.S., F.U., I.S., Y.S., M.A.W., K.E.M., E.K., N.M., T.S., G.K.L., C.H.U., S.M.B., A.D.I., B.P., R.A.J., H.G., S.R., A.L.R., C.G., K.I., L.K.C., D.C.K., T.M.M., S.E.H., D.V.F.A., H.N., A.F.L., H.D., P.L., T.M., D.M., H.K.E., D.P., J.E.P., N.C.L., N.V., E.L.H., D.B.G., C.M., J-M.S.A., N.A.A-S., M.Z., S.S., R.A., T.S., M.S.Z., G.M.H.A-S., M.S.A-H., L.A., F.S.A., H.D., A.L., P.B., G.Z., E.A., F.Z., S.E., D.G., M.C.T., R.Y., B.P-D., A-C.G., E.V., V.C-E., A.V.M., A.H., D.T., S.N.W., V.S.A., J.A.R., V.T., S.O., J.R.L., H.H., W.C.E., E.E.D., A.A.J., J.L.; Methodology: A.V.D., M.A.A., S.K., F.U., I.S., M.A.W., E.K., V.T., S.N.W., S.O., A.A.J., E.E.D., J.L.; Writing – original draft: A.V.D., M.A.A., S.K., E.K., S.N.W., E.E.D., A.A.J., J.L.; Writing – review & editing: A.V.D., M.A.A., S.K., E.E.D., A.A.J., J.L., S.N.W., J.R.L., W.C.E., V.T.

## Competing interests

BCM and Miraca Holdings have formed a joint venture with shared ownership and governance of Baylor Genetics (BG), which performs clinical microarray analysis (CMA), clinical ES (cES), and clinical biochemical studies. JRL serves on the Scientific Advisory Board of the BG. The Department of Molecular and Human Genetics at Baylor College of Medicine receives revenue from clinical genetic testing conducted at BG Laboratories. JRL has stock ownership in 23andMe, is a paid consultant for Genomics International, and is a coinventor on multiple United States and European patents related to molecular diagnostics for inherited neuropathies, eye diseases, genomic disorders, and bacterial genomic fingerprinting. DP provides consulting service for Ionis Pharmaceuticals. The other authors declare no competing interests.

## Additional information

Avinash V. Dharmadhikari [1,2,67], Maria Alba Abad [3,67], Sheraz Khan [4,5,6,7,67], Reza Maroofian [8], Tristan T. Sands [9], Farid Ullah [4,5], Itaru Samejima [3], Yanwen Shen [10,11,12,13], Martin A. Wear [14], Kiara E. Moore [4,5], Elena Kondakova [15], Natalia Mitina [15], Theres Schaub [16], Grace K. Lee [17], Christine H. Umandap [18,19], Sara M. Berger [19], Alejandro D. Iglesias [19], Bernt Popp [20], Rami Abou Jamra [20], Heinz Gabriel [21], Stefan Rentas [22], Alyssa L. Rippert [23], Christopher Gray [23], Kosuke Izumi [23], Laura K. Conlin [24], Daniel C. Koboldt [25,26], Theresa Mihalic Mosher [27], Scott E. Hickey [26,28], Dara V. F. Albert [26,29], Haley Norwood [30], Amy Feldman Lewanda [31], Hongzheng Dai [32,33], Pengfei Liu [32,33], Tadahiro Mitani [32], Dana Marafi [32,34], Hatice Koçak Eker [35], Davut Pehlivan [32,36,37], Jennifer E. Posey [32], Natalie C. Lippa [38], Natalie Vena [38], Erin L. Heinzen [39,40], David B. Goldstein [41], Cyril Mignot [42], Jean-Madeleine de Sainte Agathe [43], Nouriya Abbas Al-Sannaa [44], Mina Zamani [45,46], Saeid Sadeghian [47], Reza Azizimalamiri [47], Tahere Seifia [45,46], Maha S. Zaki [48], Ghada M. H. Abdel-Salam [48], Mohamed S. Abdel-Hamid [49], Lama Alabdi [50], Fowzan Sami Alkuraya [50], Heba Dawoud [51], Aya Lofty [51], Peter Bauer [52], Giovanni Zifarelli [52], Erum Afzal [53], Faisal Zafar [53], Stephanie Efthymiou [8], Daniel Gossett [54,55], Meghan C. Towne [27], Raey Yeneabat [56], Belen Perez-Duenas [57,58,59], Ana Cazurro-Gutierrez [58,59], Edgard Verdura [58,60], Veronica Cantarin-Extremera [61,62], Ana do Vale Marques [63], Aleksandra Helwak [3], David Tollervey [3], Sandeep N. Wontakal [56], Vimla S. Aggarwal [64], Jill A. Rosenfeld [32], Victor Tarabykin [15,16], Shinya Ohta [65], James R. Lupski [32,36,66], Henry Houlden [8], William C. Earnshaw [3], Erica E. Davis [4,5] ✉, A. Arockia Jeyaprakash [3,60] ✉ & Jun Liao [64] ✉

[1]Department of Pathology and Laboratory Medicine, Children's Hospital Los Angeles, Los Angeles, CA 90027, USA. [2]Keck School of Medicine, University of Southern California, Los Angeles, CA 90033, USA. [3]Institute of Cell Biology, University of Edinburgh, Edinburgh, United Kingdom. [4]Stanley Manne Children's Research Institute, Ann & Robert H. Lurie Children's Hospital of Chicago, Chicago, IL 60611, USA. [5]Departments of Pediatrics and Cell and Developmental Biology, Feinberg School of Medicine, Northwestern University, Chicago, IL 60611, USA. [6]Human Molecular Genetics Lab, Health Biotechnology Division, National Institute for Biotechnology and Genetic Engineering (NIBGE-C), Faisalabad, Pakistan. [7]Pakistan Institute of Engineering and Applied Sciences (PIEAS), Islamabad, Pakistan. [8]Department of Neuromuscular Diseases, University College London, Queen Square, Institute of Neurology, WC1N 3BG London, UK. [9]Department of Neurology, Vagelos College of Physicians and Surgeons, Columbia University, New York, NY 10032, USA. [10]Translational Research Center for the Nervous System, Brain Cognition and Brain Disease Institute (BCBDI), Shenzhen Institute of Advanced Technology, Chinese Academy of Sciences, 518055 Shenzhen, Guangdong, China. [11]Faculty of Life and Health sciences, Shenzhen University of Advanced Technology, 518055 Shenzhen, Guangdong, China. [12]Department of Pediatrics, Chinese PLA General Hospital, Medical School of Chinese People's Liberation Army, 100853 Beijing, China. [13]Department of Pediatrics, Fujian Medical University Union Hospital, 350001 Fuzhou, China. [14]Edinburgh Protein Production Facility (EPPF), University of Edinburgh, King's Buildings, Max Born Crescent, Edinburgh EH9 3BF, UK. [15]Institute of Neuroscience, Laboratory of Genetics of Brain Development, National Research Lobachevsky State University of Nizhny Novgorod, 60302223 Gagarin avenue, Nizhny, Novgorod, Russia. [16]Institute of Cell and Neurobiology, Charité Universitätsmedizin Berlin, 10117 BerlinCharitéplatz 1Germany. [17]Personalized Care (PCARE) Program, Department of Pathology and Laboratory Medicine; The Saban Research Institute, Children's Hospital Los Angeles, Los Angeles, CA 90027, USA. [18]Medical Genetics, DMG Children's Rehabilitative Services, Phoenix,

AZ 85013, USA. [19]Division of Clinical Genetics, Department of Pediatrics, Columbia University, Vagelos College of Physicians and Surgeons, New York, NY, USA. [20]Institute of Human Genetics, University of Leipzig Medical Center, Leipzig, Germany. [21]Praxisfür Humangenetik Tübingen, Tübingen, Germany. [22]Department of Pathology, Duke University School of Medicine, Durham, NC, USA. [23]Division of Human Genetics, Department of Pediatrics, Children's Hospital of Philadelphia, Philadelphia, PA, USA. [24]Division of Genomic Diagnostics, Department of Pathology and Laboratory Medicine, Children's Hospital of Philadelphia, Philadelphia, PA, USA. [25]The Steve and Cindy Rasmussen Institute for Genomic Medicine, Nationwide Children's Hospital, Columbus, OH, USA. [26]Department of Pediatrics, The Ohio State University College of Medicine, Columbus, OH, USA. [27]Ambry Genetics, Aliso Viejo, CA, USA. [28]Division of Genetic & Genomic Medicine, Nationwide Children's Hospital, Columbus OH 43205 OH, USA. [29]Division of Neurology, Nationwide Children's Hospital, Columbus OH 43205 OH, USA. [30]GeneDx, LLC., Gaithersburg, MD, USA. [31]Rare Disease Institute, Children's National Hospital, Washington, DC, USA. [32]Department of Molecular and Human Genetics, Baylor College of Medicine, Houston, TX, USA. [33]Baylor Genetics, Houston, TX, USA. [34]Department of Pediatrics, Faculty of Medicine, Kuwait University, Safat, Kuwait. [35]Department of Medical Genetics, Konya City Hospital, Konya, Turkey. [36]Texas Children's Hospital, Houston, TX, USA. [37]Section of Pediatric Neurology and Developmental Neuroscience, Department of Pediatrics, Baylor College of Medicine, Houston, TX, USA. [38]Department of Medicine, Columbia University Irving Medical Center, New York, NY 10032, USA. [39]Division of Pharmacotherapy and Experimental Therapeutics, Eshelman School of Pharmacy, University of North Carolina, Chapel Hill, NC, USA. [40]Department of Genetics, School of Medicine, University of North Carolina, Chapel Hill, NC, USA. [41]Institute for Genomic Medicine, Columbia University Irving Medical Center, New York, NY 10032, USA. [42]Département de Génétique, APHP Sorbonne Université, 75013 Paris, France. [43]Department of Medical Genetics, Groupe Hospitalo-Universitaire Pitié-Salpêtrière, AP-HP.Sorbonne Université, Paris, France. [44]Pediatric Services, John Hopkins Aramco Health Care, Dhahran, Saudi Arabia. [45]Department of Biology, Faculty of Science, Shahid Chamran University of Ahvaz, Ahvaz, Iran. [46]Narges Medical Genetics and Prenatal Diagnosis Laboratory, Kianpars, Ahvaz, Iran. [47]Department of Pediatric Neurology, Golestan Medical, Educational, and Research Center, Ahvaz Jundishapur University of Medical Sciences, Ahvaz, Iran. [48]Clinical Genetics Department, Human Genetics and Genome Research Institute, National Research Centre, 12622 Cairo, Egypt. [49]Medical Molecular Genetics Department, Human Genetics and Genome Research Institute, National Research Centre, 12622 Cairo, Egypt. [50]Department of Translational Genomics, Center for Genomic Medicine, King Faisal Specialist Hospital and Research Center, Riyadh, Saudi Arabia. [51]Pediatrics Department, Faculty of Medicine, Tanta University, El-Geesh Street, Tanta 31527, Egypt. [52]CENTOGENE GmbH, Am Strande 7, 18055 Rostock, Germany. [53]Department of Development Pediatrics, The Children's Hospital and The Institute of Child Health, Multan, Pakistan. [54]Texas Child Neurology, Plano, TX 75024, USA. [55]Neurology Consultants of Dallas, Dallas, TX 75243, USA. [56]Departments of Pathology and Genetic Medicine, Johns Hopkins University School of Medicine, Baltimore, MD 21205, USA. [57]Department of Paediatric Neurology, Hospital Vall d'Hebron, Barcelona, Spain. [58]Vall d'Hebron Research Institute, Barcelona, Spain. [59]Department of Paediatrics, Universitat Autònoma de Barcelona, Barcelona, Spain. [60]Molecular Biology CORE, Biomedical Diagnostic Center (CDB), Hospital, l Clínic de Barcelona, Barcelona, Spain. [61]Department of Paediatric Neurology, Hospital Infantil Niño Jesús, Madrid, Spain. [62]Centro de Investigación Biomédica en Red de Enfermedades Raras (CIBERER (GCV23/ER/3)), ISCIII, Madrid, Spain. [63]Gene Center, Department of Biochemistry, Ludwig-Maximilians Universität, Munich, Germany. [64]Department of Pathology and Cell Biology, Columbia University Irving Medical Center, New York, NY 10032, USA. [65]Institute for Genetic Medicine Pathophysiology, Hokkaido University, Sapporo, Japan. [66]Department of Pediatrics, Baylor College of Medicine, Houston, TX, USA. [67]These authors contributed equally: Avinash V. Dharmadhikari, Maria Alba Abad, Sheraz Khan. ✉e-mail: eridavis@luriechildrens.org; jeyaprakash.arulanandam@ed.ac.uk; jparul@genzentrum.lmu.de; jl5098@cumc.columbia.edu

