## [Transparent Peer Review file · Nature Communications]

RNA methyltransferase SPOUT1/CENP-32 links mitotic spindle organization with the neurodevelopmental disorder SpADMiSS

Corresponding Author: Dr Jun Liao

Version 0:

Reviewer comments:

Reviewer #1

(Remarks to the Author)

The authors identified multiple bi-allelic SPOUT1/CENP-32 variants associated with neurodevelopmental disorders. By combining SPOUT1/CENP-32 knockout zebrafish modeling, crystal structure analysis, and methyltransferase activity assays, the authors have successfully demonstrated that SPOUT1/CENP-32 missense variants are pathogenic, possibly due to misassembly of the spindle during mitosis and impaired methyltransferase activity. The authors' main conclusions are supported by several different well-controlled experiments. Overall, the findings are novel and important for the SPOUT1/CENP-32 associated neurodevelopmental disorders. My main concerns, which could be addressed by additional experiments or analysis, are as follows.

Major points:

1. The authors showed that all affected individuals had bi-allelic variants with 10/18 families segregating homozygous variants whereas 8/18 pedigrees had compound heterozygous variants. Bi-allelic variants are relatively rare in neurodevelopmental disorders. It will be valuable for the authors to perform some analyses on people carrying only one copy of SPOUT1/CENP-32 variants. In particular, it will be interesting to confirm whether the parents and siblings from the 18 families display any similar symptoms (Figure S1). This is clinically important, especially for those people who only carry only one copy of the variants. This is because in many cases NDD patients can have neurodevelopmental defects with heterozygous mutations. It is important to make this clear for SPOUT1/CENP-32 variants.
2. The study highlights haploinsufficiency in enzyme activity in both homozygous mutant variants in patients and heterozygous knockout fish. The authors showed that all homozygous knockout zebrafish are lethal. What about the heterozygous knockout fish? Do they show any kind of behavioral abnormalities? It will be valuable to perform some behavioral tests, e.g. novelty tank test, mirror test, social interaction test, shoaling test, learning behavior etc. Importantly, do these fish show behavioral phenotypes similar to those seen in patients?
3. The authors showed that SPOUT1/CENP-32 may form dimers which is essential for MTase activity (Figure 5). In the zebrafish in vivo complementation assays, the effects of the 11 variants showed different rescue effects (Figure 3), suggesting that they may function differently (Figure 7A), i.e. disrupt dimerisation, impair substrate binding. Is it possible to test these possibilities? Further experiments could provide deeper insights into the mechanisms underlying the observed pathogenicity.
4. It has been shown that CENP-32 is required for centrosomes to behave like spindle poles capable of nucleating both spindle and astral microtubules (Shinya Ohta et al., 2015). Here the authors emphasized that the methyltransferase activity of SPOUT1/CENP-32 is crucial for proper spindle assembly. However, SPOUT1/CENP-32 may have dual roles in spindle assembly and methyltransferase activity. Is it possible to dissociate the two roles? In addition, it is important to clarify whether the spindle assembly function is dependent on the methyltransferase activity.
5. To dissociate the role of methyltransferase activity, conditional knockout animal models may be useful. Ideally, postnatal deletion of Spout1/Cenp-32 in neurons, i.e. in mice, would bypass the misassembly of the spindle during mitosis and study the contribution of methyltransferase activity in brain development and function.

Minor points:

Line 306: $q < 0.0001$ for *spout1/cenp-32*^{-/-} vs *spout1/cenp-32*^{+/+}

It should be $p < 0.0001$.

Please also correct similar typos throughout the manuscript.

Line 1571: Omit the word "were" from the sentence "in vitro were Survivin protein was used as a negative control".

Reviewer #2

(Remarks to the Author)

The manuscript by Dharmadhikari et al. reports on the finding of biallelic SPOUT1 variants causing a novel syndrome. The authors provide detailed phenotypic descriptions and accurate genetic data from 18 families with 24 affected individuals. Knockout of *spout1* in zebrafish larvae revealed features from the patients (i.e. microcephaly) and patient variants failed to rescue a Morpholino-knockdown phenotype. Further in vitro studies shed light on the function of SPOUT1 as an active RNA methyltransferase that is required for maintaining spindle integrity. Authors present data supporting the hypothesis that patient variants lead to a reduced enzymatic activity of SPOUT1.

The study and manuscript are of good quality and provide relevant content for a variety of readership, from clinicians to molecular scientists.

I have major concerns regarding the manuscript:

1) The zebrafish data is overall convincing, however for Morpholino phenotypes it is necessary to compare to other controls than uninjected (i.e. Control Morpholino injected larvae). As shown in Fig 2C and Fig S9 both, stable knockout and Morpholino-knockdown, reduce significantly the body length of the larvae and the head size should be normalized for this general growth effect. This is important since the pure manipulation of injection of the embryos causes inflammation and often leads to a developmental delay. line 389 "We did not detect marked differences in body length phenotype ..." should be adjusted accordingly.

2) It seems that the animal models are more severe than the patients with biallelic SPOUT1 variants, since the embryos and larvae are not viable (mice, zebrafish and *C. elegans*). The authors should address the topic "viability/mortality" in more details. For example it would be interesting if the Mo-knockdown showed an dose-dependent effect. A Kaplan-Meier plot would be helpful. Mortality and stillbirths in the families should be commented in the manuscript.

3) With such a widespread mechanism as mitotic spindle organization being affected, how do the authors explain the specific NDD phenotype. Why do the patients do not present with phenotypes in other tissues/organs with high mitotic activity (e.g. bone marrow, gastrointestinal, etc.)?

Minor concerns:

1) When weight, size and head measurements were reported, please add percentiles or (better) z-scores or SD values

- Line 263: should be "spasms"

- Fig 4: "pink rectangles" are actually red

- Readability of legends and text in the figures in general could be improved (e.g. Fig 1)

Reviewer #3

(Remarks to the Author)

The manuscript "RNA methyltransferase SPOUT1/CENP-32 links mitotic spindle organization with the neurodevelopmental disorder SpADMiSS" (NCOMMS-24-06474-T) by Dharmadhikari et al. attempts to elucidate the function of a previously identified putative SPOUT RNA methyltransferase SPOUT1/CENP-32. The authors identified 24 individuals who have biallelic variants in SPOUT1/CENP-32 with various missense mutations. Phenotypical analysis of zebrafish *spout1/cenp-32* mutants derived from the patients recaptured the features showing reduced head sizes with concomitant apoptosis.

Structural and biochemical analysis of SPOUT1/CENP-32 revealed the impacts of disease-associated mutations in the catalytic domain. In this paper, the authors discovered that SPOUT1/CENP-32 has enzymatic activity towards RNA for the first time. The authors also found that RNA methyltransferase ability of the SPOUT1/CENP-32 is necessary to maintain spindle integrity. At this stage, the authors have comprehensively elucidated the functions of SPOUT1/CENP-32 through an in-depth analysis from genetic approaches to structural data.

The overall impression of the article is good, as this is the first paper to show that reduced RNA methylation activity of SPOUT1/CENP-32 is associated with neurodevelopmental abnormalities. However, further experiments are needed to clarify certain points.

Major comments:

1. The authors state in lines 389-391 that there are no obvious differences in body length, but in fact there are statistically significant differences (Fig. 2C, Fig. S4E, and Fig. S9B). Why is this? This should be considered in the context of the pathology in humans.

2. There is a problem with the experiment measuring the proportion of apoptosis and cell cycle arrest in the head region of zebrafish larvae. The measurement range is completely different in each figure (Fig. 4D, F and Fig. S6D, F). Although the authors described as "consistently defined head region" in the methods section, I request detailed explanation of the basis for defining the measurement range in each experiment.

3. The expression of N86D, G98S, T353M mutations in SPOUT1/CENP-32 seems to cause chromosome segregation errors

even in the control of siRNA (Fig. 8B, C). Also, a significant decrease in cell survival is observed in the T353M mutation (Fig. 8D). In other words, could these mutants be dominant negative? Do these heterozygous individuals of these mutants not develop pathology? To verify this, the lifespan of zebrafish heterozygotes with the T353M mutation should be measured.

4. The authors should quantify the expression level of Dox induced SPOUT1/CENP-32 variants to ensure comparable expressions in each cell line (Fig. S13).
5. The authors have demonstrated through various approaches that the RNA methyltransferase activity of SPOUT1/CENP-32 is important for centrosome tethering to spindle poles. However, as the authors discuss in the manuscript, its actual mechanism is unclear, and this is the most important point. The authors refer that WDR8 and DCNT4 interact with SPOUT1/CENP-32 homologue at the centrosome and are involved in spindle formation in fission yeast. This can be verified with a simple epistasis analysis using fission yeast. While this may not be necessary for this paper, I suggest trying this experiment to support that idea.
6. As for the structures of SPOUT1/CENP-32 bound to co-factors, the authors described that SAH was identified in the cofactor binding site, despite the absence of supplementation with either SAM or SAH. It remains unclear how the authors determined the electron density in the cofactor binding site to be that of SAH, rather than SAM. Although Figure S13 B and C show the electron densities for SAH and SAM in each SPOUT1/CENP-32 structure, the differences between them are not clearly defined. The authors are advised to provide detailed views of SAH and SAM, overlaying each with their respective electron density maps in the SPOUT1/CENP-32 structures.

Minor comments:

1. Line 342: "S6A, C" should be "S6A-C".
2. Lines 355-357: Please explain briefly what TUNEL and pHH3 (histone H3-S10ph) are as markers.
3. Line 968: "Hepes" should be "HEPES".
4. Fig. 2D: Zebrafish genotyping verification data should be shown in supplementary figure.
5. Fig. 3 and 4 should be inverted as the authors describe Fig. 4 prior to Fig. 3 on the original manuscript.
6. Fig. 6C-E and Fig. 8: Please add the explanation what "C" is in the legends (control?).
7. Fig. S3B and S3D: Please add "bp" in the figure.
8. Fig. S10: The plots should be separately placed with the same category (Head area and Body length).

Version 1:

Reviewer comments:

Reviewer #1

(Remarks to the Author)

In the revised version, the authors added more analyses or discussion, which improved the manuscript. I still have two more questions.

1. The authors claim that the heterozygous mutations in humans do not lead to any kind of deficits, but the heterozygous mutant in zebrafish leads to severe deficits in head size and spindle assembly. What causes the differences? This notion should be carefully discussed in the revision. Since different mutations lead to minor to severe deficits in zebrafish, are the differences due to the heterogeneity of human patients? Is it possible to compare the deficits of zebrafish lines and human patients carrying the same mutations?

2. The authors show in Figure S5 that the heterozygous larval fish display reduced locomotion. Clinical patients have showed developmental delay, intellectual disability, and seizures. The purpose of disease modeling is to test the causal relationship between mutations and clinical symptoms, and to explore the possible mechanisms/drug targets. It is very important to evaluate whether Spout1 haploinsufficiency indeed leads to these deficits. Here the DD phenotype is evidenced by the reduced body length and head size. Since the heterozygous fish are not lethal, other phenotypes should be measured in these adult fish, which will strengthen the conclusion of this manuscript.

Reviewer #2

(Remarks to the Author)

The authors have appropriately addressed my comments, and the revised manuscript has markedly improved.

Reviewer #3

(Remarks to the Author)

The authors have responded appropriately to most of my comments, and for the points they could not answer, they give convincing reasons. The actual molecular mechanism of SPOUT1/CENP-32 in the centrosome is not yet clear, but I don't think it is essential for this article, therefore it should not hinder the publication. In conclusion, I am satisfied with the revisions and recommend publication of this manuscript in Nature Communications. Please see the minor comments below to improve the quality of the manuscript.

1. Regarding the significant differences in fish body length (Fig. 2C, Fig. S4E, and Fig. S9B), I do not disagree with the authors' interpretation that the effect size is modest given the sample size. However, although the authors mention that they have revised the main text accordingly, I was unable to find this comment. Perhaps it would be better to insert it on line 340 in the revised manuscript. Please check and consider where best to place this comment.

2. Lines 1014, 1026 and 1116: “HEPESHEPES” should be corrected to “HEPES”.

Below are point-by-point responses to the reviewer comments. Additional revisions made to the manuscript are highlighted as track changes.

REVIEWER COMMENTS

Reviewer #1 (Remarks to the Author):

The authors identified multiple bi-allelic SPOUT1/CENP-32 variants associated with neurodevelopmental disorders. By combining SPOUT1/CENP-32 knockout zebrafish modeling, crystal structure analysis, and methyltransferase activity assays, the authors have successfully demonstrated that SPOUT1/CENP-32 missense variants are pathogenic, possibly due to misassembly of the spindle during mitosis and impaired methyltransferase activity. The authors' main conclusions are supported by several different well-controlled experiments. Overall, the findings are novel and important for the SPOUT1/CENP-32 associated neurodevelopmental disorders. My main concerns, which could be addressed by additional experiments or analysis, are as follows.

Major points:

1. The authors showed that all affected individuals had bi-allelic variants with 10/18 families segregating homozygous variants whereas 8/18 pedigrees had compound heterozygous variants. Bi-allelic variants are relatively rare in neurodevelopmental disorders. It will be valuable for the authors to perform some analyses on people carrying only one copy of SPOUT1/CENP-32 variants. In particular, it will be interesting to confirm whether the parents and siblings from the 18 families display any similar symptoms (Figure S1). This is clinically important, especially for those people who only carry only one copy of the variants. This is because in many cases NDD patients can have neurodevelopmental defects with heterozygous mutations. It is important to make this clear for SPOUT1/CENP-32 variants.

We agree with the reviewer that NDD patients are more likely to have a dominant inheritance pattern. However, it is estimated that autosomal recessive inheritance contributes to about 10% of neurodevelopmental cases in an outbred population, and in consanguineous families, this is estimated to be an order of magnitude higher (PMID: 30459488). Specifically, when focusing on genes with similar cellular functions, the inheritance pattern for other genes involved in centrosome-based disorders: CDK5RAP2 (MIM: 608201), PCNT (MIM: 605925), WDR62 (MIM: 613583), and ASPM (MIM: 605481) is also autosomal recessive. Consistent with an autosomal recessive inheritance pattern, heterozygous carrier parents in the families described in this cohort are reportedly normal and unaffected, with no neurodevelopmental phenotypes encountered as part of the clinical work-up. Additionally, the variants reported in this study have also been found at a low frequency in the gnomAD v4.0.0 database (Table S2), and all of them are only seen in the heterozygous state. This demonstrates that heterozygotes can be found in the general population. For clarity, we have now added this statement to the results of the manuscript (please see page 8 line 311; manuscript file with track changes).

2. The study highlights haploinsufficiency in enzyme activity in both homozygous mutant variants in patients and heterozygous knockout fish. The authors showed that all homozygous knockout zebrafish are lethal. What about the heterozygous knockout fish? Do they show any kind of behavioral abnormalities? It will be valuable to perform some behavioral tests, e.g. novelty tank test, mirror test, social interaction test, shoaling test, learning behavior etc. Importantly, do these fish show behavioral phenotypes similar to those seen in patients?

We thank the reviewer for raising this question. As suggested, we performed preliminary behavioral tracking of all three genotypes with the DanioVision platform and observed a reduced mean activity in +/- versus +/+ (25%). These data are now presented in revised Figure S5A, B. Further work beyond the scope of this manuscript will be required to understand the molecular basis for this difference and whether it relates to SPOUT1 heterozygous carrier humans, particularly since we did not observe any significant differences between these two genotypes for head size (bright field), neuroanatomy, (acetylated tubulin), cell death (TUNEL) and cell cycle progression (phospho-Histone-H3). Extrapolating these larval phenotypes to human patient phenotypes would be overly speculative, particularly since the majority of human variants are missense changes that may have some minimal residual function, and the zebrafish mutation is a frameshifting putative loss of function allele.

3. The authors showed that SPOUT1/CENP-32 may form dimers which is essential for MTase activity (Figure 5). In the zebrafish in vivo complementation assays, the effects of the 11 variants showed different rescue effects (Figure 3), suggesting that they may function differently (Figure 7A), i.e. disrupt dimerisation, impair substrate binding. Is it possible to test these possibilities? Further experiments could provide deeper insights into the mechanisms underlying the observed pathogenicity.

We thank the reviewer for this suggestion. Amongst the different SPOUT1/CENP-32 variants tested in this study, G98S is located at the dimeric interface, G244S is located in the OB-fold, and the rest are located in the loops that form the cofactor binding pocket. As none of these variants completely abolishes the catalytic activity, we speculated that these variants only subtly affect SPOUT1/CENP-32 structure and/or cofactor/RNA substrate binding. Although precisely teasing apart the molecular basis for the differential effects of SPOUT1/CENP-32 variants is beyond the scope of this manuscript, we characterized the effect of variants on their ability to dimerize and RNA substrate binding using Size Exclusion Chromatography (SEC) and Mass Photometry (MP) experiments and Electrophoretic Mobility Shift Assays (EMSA) assays, respectively. As we expected, only G98S showed a tendency to form smaller species (as evidenced by the shift towards the right of the main peak on the chromatographic profile), while other variants eluted at an elution volume nearly identical to that of SPOUT1/CENP-32 WT. Further characterization of the variants using Mass Photometry showed that at a nanomolar concentration, G98S existed predominantly as a monomer whilst the other variants remained as dimers.

Unlike the SEC and MP analyses, the results of the EMSA assays we carried out to analyze the effect of variants on RNA binding were unfortunately not straightforward to interpret. While SPOUT1/CENP-32 N86D and SPOUT1/CENP-32 G98S variants showed apparently similar RNA binding as SPOUT1/CENP-32 WT, the electrophoretic migration pattern for SPOUT1/CENP-32 G244S, T289M, G293S and T353M variants were strikingly different from SPOUT1/CENP-32 WT with RNA showing a sub-shift, suggesting a conformational change in the structure of the RNA, towards a more compact form, upon SPOUT1/CENP-32 binding. The sub-shift could be due to perturbed cofactor and RNA binding activity, but further analyses should be performed to assess this unambiguously. Thus, as the data is inconclusive, we decided not to include the results of the EMSAs in the main manuscript, and we only show them here in response to reviewers (please find the figure below, blue arrows point to the sub-shift).

However, the SEC and MP data have now been included in new Figure S16 and we have now added the following statement to the MS_track changes on page 17 line 593 ‘Size exclusion chromatography and mass photometry experiments show that all variants except SPOUT1/CENP-32 G98S exist as dimers (Figure S16A-F). Considering that the loops forming the cofactor binding site are stabilized by the dimeric interface and all variants except G98S are either near the cofactor binding site or within the OB-fold domain implicated in nucleic acid binding, the cofactor and/or substrate binding and turnover of the SPOUT1/CENP-32 variants may be affected, making them less efficient enzymes in affected individuals.’

4. It has been shown that CENP-32 is required for centrosomes to behave like spindle poles capable of nucleating both spindle and astral microtubules (Shinya Ohta et al., 2015). Here the authors emphasized that the methyltransferase activity of SPOUT1/CENP-32 is crucial for proper spindle assembly. However, SPOUT1/CENP-32 may have dual roles in spindle assembly and methyltransferase activity. Is it possible to dissociate the two roles? In addition, it is important to clarify whether the spindle assembly function is dependent on the methyltransferase activity.

We apologize if our phrasing in this section of the manuscript was not clear. As already described in the manuscript, the catalytic site mutant we designed based on the crystal structures (Figures 5B and S14F), where we mutated the Ala356 located at the center of the SAM binding pocket to Asn (A356N), an amino acid with a longer side chain,

showed perturbed SAM binding (Figure S14A-C) and strongly reduced methyltransferase activity in the methyltransferase assays (Figure 6B). As shown in (Figure 6C and 6D), A356N mutant failed to rescue the centrosome detachment phenotype in the siRNA rescue assay, indicating that the spindle assembly function of SPOUT1/CENP-32 is dependent on its methyltransferase activity. For clarity, we have now added the following statement to the MS_track changes on page 16 line 567 'These data together indicate that the role of SPOUT1/CENP-32 in regulating spindle organisation is dependent on its methyltransferase activity.'

5. To dissociate the role of methyltransferase activity, conditional knockout animal models may be useful. Ideally, postnatal deletion of Spout1/Cenp-32 in neurons, i.e. in mice, would bypass the misassembly of the spindle during mitosis and study the contribution of methyltransferase activity in brain development and function.

As the reviewer has correctly pointed out, Spout1 can have postmitotic function beyond its role in the mitotic spindle assembly as indicated by its expression. However, generation of conditional knockout mouse model and analysis usually takes two-three years. We agree with the reviewer that it is important to address, however, we believe it is far beyond the scope of the current manuscript.

Minor points:

Line 306: $q < 0.0001$ for spout1/cenp-32^{-/-} vs spout1/cenp-32^{+/+}

It should be $p < 0.0001$.

Please also correct similar typos throughout the manuscript.

These typos have been corrected.

Line 1571: Omit the word "were" from the sentence "in vitro were Survivin protein was used as a negative control".

We have now replaced 'were' by 'where' as follows: 'Titration of methyltransferases (SPOUT1/CENP-32 and NSUN6) to assess the methyltransferase activity of SPOUT1/CENP-32 in vitro where Survivin protein was used as a negative control.' in page 55, line 1804 in the new version of the MS_track changes.

Reviewer #2 (Remarks to the Author):

The manuscript by Dharmadhikari et al. reports on the finding of biallelic SPOUT1 variants causing a novel syndrome. The authors provide detailed phenotypic descriptions and accurate genetic data from 18 families with 24 affected individuals. Knockout of spout1 in zebrafish larvae revealed features from the patients (i.e. microcephaly) and patient variants failed to rescue a Morpholino-knockdown phenotype.

Further in vitro studies shed light on the function of SPOUT1 as an active RNA methyltransferase that is required for maintaining spindle integrity. Authors present data supporting the hypothesis that patient variants lead to a reduced enzymatic activity of SPOUT1.

The study and manuscript are of good quality and provide relevant content for a variety of readership, from clinicians to molecular scientists.

I have major concerns regarding the manuscript:

1) The zebrafish data is overall convincing, however for Morpholino phenotypes it is necessary to compare to other controls than uninjected (i.e. Control Morpholino injected larvae). As shown in Fig 2C and Fig S9 both, stable knockout and Morpholino-knockdown, reduce significantly the body length of the larvae and the head size should be normalized for this general growth effect. This is important since the pure manipulation of injection of the embryos causes inflammation and often leads to a developmental delay. line 389 “We did not detect marked differences in body length phenotype ...” should be adjusted accordingly.

We have improved the scientific rigor of the morpholino studies thanks to the reviewer's suggestion. We include new data showing no significant differences between zebrafish larvae injected with sham (phenol red), standard control morpholino or 9ng spout1/cenp-32 MO (please see Figure S6C, D). We would like to point out to the reviewer that although significant, the effect size on body length is modest (<3%). We are cautious in over interpreting these body length data and have revised the main text accordingly.

2) It seems that the animal models are more severe than the patients with biallelic SPOUT1 variants, since the embryos and larvae are not viable (mice, zebrafish and C. elegans). The authors should address the topic “viability/mortality” in more details. For example it would be interesting if the Mo-knockdown showed an dose-dependent effect. A Kaplan-Meier plot would be helpful.

Thank you, we agree. We remind the reviewer that transient suppression using a MO to target spout1/cenp-32 results in a dose-dependent effect; please see revised Figure S6. We agree that a Kaplan-Meier plot for spout1/cenp-32 mutant zebrafish is helpful; please see these new data in revised Figure S5C. Additionally, it is important to note that the majority of human variants are missense changes that may have some minimal residual function, and the zebrafish mutation is a frameshifting putative loss of function allele.

Mortality and stillbirths in the families should be commented in the manuscript.

Mortality and stillbirths observed in the described families in this cohort appear to be comparable to that expected in the general population. However, due to the relatively small sample size of our cohort, further studies with more families will be needed in the future. We have added this statement to the manuscript_track changes (page 8 line 313), as the reviewed suggested.

3) With such a widespread mechanism as mitotic spindle organization being affected, how do the authors explain the specific NDD phenotype. Why do the patients do not present with phenotypes in other tissues/organs with high mitotic activity (e.g. bone marrow, gastrointestinal, etc.)?

*We would like to thank the reviewer for pointing this out. Similar observations (the specific NDD phenotype) have also been reported in other “centrosome-based diseases”, such as CDK5RAP2, PCNT, WDR62, and ASPM, **indicating the brain is particularly susceptible to defects affecting centrosome and mitotic spindle assembly and function.** Although the biological mechanisms for this phenomenon remain a major open question in the field, several possible reasons have been proposed, including the short time window of neurogenesis, tissue-specific isoform expression, different downstream cellular and molecular pathways, and neural-specific characteristics such as polarization (PMIDs: 34862179, 37443841). We have added these possible explanations to our manuscript_track changes (please see page 5, line 225).*

Minor concerns:

1) When weight, size and head measurements were reported, please add percentiles or (better) z-scores or SD values

Percentiles have been added in Table S1.

- Line 263: should be “spasms”

Corrected.

- Fig 4: “pink rectangles” are actually red

Corrected.

- Readability of legends and text in the figures in general could be improved (e.g. Fig 1)

Figures 1A, B have been updated to improve readability.

Reviewer #3 (Remarks to the Author):

The manuscript “RNA methyltransferase SPOUT1/CENP-32 links mitotic spindle organization with the neurodevelopmental disorder SpADMiSS” (NCOMMS-24-06474-T) by Dharmadhikari et al. attempts to elucidate the function of a previously identified putative SPOUT RNA methyltransferase SPOUT1/CENP-32. The authors identified 24 individuals who have bi-allelic variants in SPOUT1/CENP-32 with various missense mutations. Phenotypical analysis of zebrafish *spout1/cenp-32* mutants derived from the patients recaptured the features showing reduced head sizes with concomitant apoptosis. Structural and biochemical analysis of SPOUT1/CENP-32 revealed the impacts of disease-associated mutations in the catalytic domain. In this paper, the authors discovered that SPOUT1/CENP-32 has enzymatic activity towards RNA for the first time. The authors also found that RNA methyltransferase ability of the SPOUT1/CENP-32 is necessary to maintain spindle integrity. At this stage, the authors have comprehensively elucidated the functions of SPOUT1/CENP-32 through an in-depth analysis from genetic approaches to structural data.

The overall impression of the article is good, as this is the first paper to show that reduced RNA methylation activity of SPOUT1/CENP-32 is associated with neurodevelopmental abnormalities. However, further experiments are needed to clarify certain points.

Major comments:

1. The authors state in lines 389-391 that there are no obvious differences in body length, but in fact there are statistically significant differences (Fig. 2C, Fig. S4E, and Fig. S9B). Why is this? This should be considered in the context of the pathology in humans.

We are grateful to the reviewer for pointing this out. The differences are statistically significant because we include so many animals in our assay, however the effect size is modest (<3%). We are cautious in over interpreting these body length data and have revised the main text accordingly.

2. There is a problem with the experiment measuring the proportion of apoptosis and cell cycle arrest in the head region of zebrafish larvae. The measurement range is completely different in each figure (Fig. 4D, F and Fig. S6D, F). Although the authors described as “consistently defined head region” in the methods section, I request detailed explanation of the basis for defining the measurement range in each experiment.

We apologize for this inconsistency. We have remeasured the images that comprise the revised Figure S7 TUNEL and pHH3 data (previously Figure S6) using a region of interest (ROI) that matches the dimensions of the ROI in Figure 4D, F. The overall conclusions remain the same, but we have replaced Figure S7D-G and information in Table S4 is updated.

3. The expression of N86D, G98S, T353M mutations in SPOUT1/CENP-32 seems to cause chromosome segregation errors even in the control of siRNA (Fig. 8B, C). Also, a significant decrease in cell survival is observed in the T353M mutation (Fig. 8D). In other words, could these mutants be dominant negative? Do these heterozygous individuals of these mutants not develop pathology? To verify this, the lifespan of zebrafish heterozygotes with the T353M mutation should be measured.

We thank the reviewer for raising this point about T353M. Our transient in vivo studies in zebrafish show no significant differences in head size and a modest effect size in body length (<2%) when the variant is ectopically expressed. Although it could be informative, generation of a T353M zebrafish mutant and study of the lifespan of heterozygotes is beyond the scope of the current work. Moreover, heterozygous carrier parents in the families described in this cohort with the N86D, G98S, and T353M variants are reportedly normal and unaffected, with no neurodevelopmental phenotypes encountered as part of the clinical work-up. In addition, all three variants in the heterozygous state are observed in populations from the gnomAD (v4.0.0) database (Table S2). Therefore, human evidence does not support a dominant negative effect for these variants. All together, we speculate that the possible dominant negative phenotype we observe in our cell-based assays is likely to be cell-type specific. The cancer-derived U2OS cells, although have been established as a good model system to study cell division, might be more sensitive to mitotic perturbations caused by the hetero-dimerization of these variants with endogenous SPOUT1/CENP-32.

4. The authors should quantify the expression level of Dox induced SPOUT1/CENP-32 variants to ensure comparable expressions in each cell line (Fig. S13).

We thank the reviewer for the suggestion. We have now quantified the expression levels of the doxycycline-induced SPOUT1/CENP-32 variants and confirmed that all the inducible cell lines show comparable expression levels. We have now added this data to new Figure S15D and S15E and added the following statement to the corresponding figure legend in the supplementary materials file: '(C-E) Representative Immunoblots of SPOUT1/CENP-32-GFP constructs (CENP-32-GFP WT, CENP-32-GFP N86D, CENP-32-GFP G98S, CENP-32-GFP G244S, CENP-32-GFP T289M, CENP-32-GFP G293S, CENP-32-GFP T353M or CENP-32-GFP A356N) and the corresponding tubulin signal as a loading control, showing comparable expression levels among all inducible cell lines. Source data are provided as a Source Data file.'

5. The authors have demonstrated through various approaches that the RNA methyltransferase activity of SPOUT1/CENP-32 is important for centrosome tethering to spindle poles. However, as the authors discuss in the manuscript, its actual mechanism is unclear, and this is the most important point. The authors refer that WDR8 and DCNT4 interact with SPOUT1/CENP-32 homologue at the

centrosome and are involved in spindle formation in fission yeast. This can be verified with a simple epistasis analysis using fission yeast. While this may not be necessary for this paper, I suggest trying this experiment to support that idea.

We thank the reviewer for this suggestion. The yeast homologue of SPOUT1/CENP-32, YMR310c, shows significant structural differences with its human counterpart. Unlike human SPOUT1/CENP-32, YMR310c does not contain an N-terminal α -helix or an OB-fold inserted into its catalytic domain, and the overall surface charge distribution is very different, with striking differences in the cofactor binding pocket (please find a figure below where we compare the crystal structures of human SPOUT1/CENP-32 71-376 and yeast YMR310c).

Moreover, YMR310c is not an essential gene in yeast.

Considering these differences, interpreting the outcome of the epistasis analysis will likely be challenging and beyond the scope of this manuscript. However, we aimed to test the dependency between SPOUT1/CENP-32 and WDR8 using the human U2OS cell line. The commercially available antibody against WDR8 (NBP1-54776, Novus Biologicals) worked for western blot but did not work for immunofluorescence studies. Thus we tested the levels of WDR8 in whole cell extracts of U2OS cells treated with control and SPOUT1/CENP-32 siRNA oligos. As you can see in the figure below, which we assembled for the response to reviewers, the western blot shows that the total levels of WDR8 do not change upon SPOUT1/CENP-32 depletion (siRNA C32) compared to the siRNA Control (siRNA C) condition. In our future studies, we will aim to study changes in WDR8, and other centrosomal proteins, from purified centrosomes of U2OS upon SPOUT1/CENP-32 depletion using mass spectrometry to understand the dependency between those proteins and further dissect the molecular mechanism of SPOUT1/CENP-32 function.

6. As for the structures of SPOUT1/CENP-32 bound to co-factors, the authors described that SAH was identified in the cofactor binding site, despite the absence of supplementation with either SAM or SAH. It remains unclear how the authors determined the electron density in the cofactor binding site to be that of SAH, rather than SAM. Although Figure S13 B and C show the electron densities for SAH and SAM in each SPOUT1/CENP-32 structure, the differences between them are not clearly defined. The authors are advised to provide detailed views of SAH and SAM, overlaying each with their respective electron density maps in the SPOUT1/CENP-32 structures.

We thank the reviewer for the suggestion. We have now included new figure panels showing the detailed views of the superposed fo-fc omit and 2fo-fc maps (Figure S14D and E, respectively) for the cofactor pockets of both SPOUT1/CENP-32 structures without (SAH-bound) and with SAM soaking (SAM-bound) with different contour levels.

The shape of the electron density surrounding the sulfur atom of the cofactor (SAH/SAM) is noticeably different in the unsoaked and SAM-soaked crystal structures. While the methyl group attached to the sulfur could be best fitted in the electron density map of the SAM-soaked crystal, the corresponding region in the unsoaked crystal (SAH bound) shows electron density surrounding the sulfur atom connected to electron density corresponding to the carbonyl oxygen of Ile351. These differences led us to model SAH in the unsoaked and SAM in the SAM-soaked crystal structures.

Together with the new panels (Figure S14D-E), we have added the following text to the corresponding figure legend in the supplementary materials file: '(D, E) Close-up view of the cofactor binding pocket of SPOUT1/CENP-32 71-376 without and with SAM soaking (shown in gold and grey, respectively) with their corresponding omit maps (fo-fc) contoured at 3 sigma (D, right panel) and at 2 sigma (D, left panel) and 2fo-fc map at 1 sigma (E). Arrows highlight differences in the electron density map observed in the structure of the SAM-soaked crystal.'

Minor comments:

1. Line 342: "S6A, C" should be "S6A-C".

Corrected.

2. Lines 355-357: Please explain briefly what TUNEL and pHH3 (histone H3-S10ph) are as markers.

Brief explanations have been added in the text for TUNEL and pHH3 staining.

3. Line 968: “Hepes” should be “HEPES”.

We have now corrected this in the text.

4. Fig. 2D: Zebrafish genotyping verification data should be shown in supplementally figure.

Thank you for this suggestion. We have added representative images of sequence chromatograms corresponding to each of the three spout1 F2 genotypes in Figure S3D.

5. Fig. 3 and 4 should be inverted as the authors describe Fig. 4 prior to Fig. 3 on the original manuscript.

We double checked, and Figure 3 (in vivo complementation assay) is described before Figure 4 (anatomical and molecular markers of CNS integrity in mutants) in both the main text and the figure order.

6. Fig. 6C-E and Fig. 8: Please add the explanation what “C” is in the legends (control?).

We have now specified the meaning of ‘C’ in the figure legends for Figures 6C, 8A, S14 and S16A.

7. Fig. S3B and S3D: Please add “bp” in the figure.

Thank you. We have updated each of Figure S3B and revised S3E by including bp on the marker sizes.

8. Fig. S10: The plots should be separately placed with the same category (Head area and Body length).

We have modified revised Figure S11 (previously Figure S10) as suggested.

Below are point-by-point responses to the reviewer comments. Additional revisions made to the manuscript are highlighted as track changes.

REVIEWER COMMENTS

Reviewer #1 (Remarks to the Author):

In the revised version, the authors added more analyses or discussion, which improved the manuscript. I still have two more questions.

1. The authors claim that the heterozygous mutations in humans do not lead to any kind of deficits, but the heterozygous mutant in zebrafish leads to severe deficits in head size and spindle assembly. What causes the differences? This notion should be carefully discussed in the revision. Since different mutations lead to minor to severe deficits in zebrafish, are the differences due to the heterogeneity of human patients? Is it possible to compare the deficits of zebrafish lines and human patients carrying the same mutations?

Thank you for raising this point, and for highlighting the need for clarification in the manuscript. We remind the reviewer of our approaches and data and ask that the reviewer consider the stable genetic model characterization separately from the variant assays of pathogenicity:

Stable Zebrafish Model. *In this model, a frameshift variant (c.543_544insA; p.R215Kfs*7) was introduced into the Zebrafish spout1/cenp-32 gene using CRISPR/Cas9 genome editing. Although the homozygous spout1 mutants show significant differences from WT in eight measurements including head size (Fig. 2A, B), body length (Fig 2A, C), optic tecta area (Fig 4A,B), number of intertectal neurons (Fig 4A, C), TUNEL positive cells in the head (Fig D, E), pHH3 positive cells in the head (Fig 4F, G), locomotion (Fig S5A, B), and survivability (Fig S5C), **the heterozygous Zebrafish mutants show no differences from WT in these measurements except for locomotion, which has an intermediate reduction.** However, why do those heterozygous parents of affected humans not display any detectable phenotypes? First, allele type/severity is one explanation. The frameshifting variant in Zebrafish is a putative loss of function (LOF). By contrast, we did not identify any affected humans with biallelic putative LOF variants (frameshifting, nonsense, or mRNA splice site) and report only two cases with one LOF and one missense variant (Table S1). We speculate that missense alleles may result in minimal residual function and that an absence of humans with biallelic LOF variants is indicative of incompatibility with life. Among the carrier parents, there are only two individuals with heterozygous LOF variants (parents of C-II-2 and R-II-1) that could be compared against heterozygous spout1 zebrafish mutants. We cannot exclude the possibility that they may have a minor subclinical phenotype, but we lack detailed neurological phenotyping information from these individuals given that they are reportedly healthy and not the focus of our study. Further explanations for variable penetrance include differences in genetic background (PMID:*

23594743), expression levels of the WT allele in trans (PMID: 18640990), compensatory alleles in cis (PMID: 26123021) and transcriptomic buffering (PMID: 26168398). Such differences are well described in the literature and could explain why a heterozygous *spout1* zebrafish mutant might display a modest locomotion phenotype or why a heterozygous *SPOUT1* human might be protected.

Zebrafish and cell based assays of variant pathogenicity. We remind the reviewer that while the *spout1* zebrafish mutant assigned physiological relevance of the gene to disease, it could not inform whether each *SPOUT1* variant in an affected human was pathogenic. For this, we performed complementation assays by depleting the endogenous transcript transiently in cells or zebrafish, and reintroduced human *SPOUT1* harboring the test variant of interest. These assays allowed us to detect statistically significant differences between abilities of WT and variants to rescue leading to conclusions of pathogenicity. A major strength of such transient assays is that they are highly tractable and enabled us to test a large allelic series (11 disease-associated variants and 1 benign control variant). We caution the reviewer that although informative, **these assays have limitations of not representing exact transcript dosage levels present in an endogenous system.** Therefore, we are reluctant to overstate differences in missense variant severity across model systems with diverse biological context and relate variant ability-to-rescue to variability in human phenotypes. We hope the reviewer agrees that the generation of genetically stable knock-in cell or animal models is beyond the scope of the manuscript.

To address these points in the manuscript, we have added the following to the end of the last paragraph of the Discussion.

*“We recognize that factors including differences in *SPOUT1/spout1* allele strength, genetic background, trans and cis factors likely contribute to differences in penetrance and expressivity observed for biallelic and heterozygous humans versus other species. Furthermore, one limitation of our study is the transient nature of our variant testing strategies in zebrafish and cells; generation and in-depth characterization of stable knock-in models will aid future mechanistic investigation.”*

2. The authors show in Figure S5 that the heterozygous larval fish display reduced locomotion. Clinical patients have showed developmental delay, intellectual disability, and seizures. The purpose of disease modeling is to test the causal relationship between mutations and clinical symptoms, and to explore the possible mechanisms/drug targets. It is very important to evaluate whether *Spout1* haploinsufficiency indeed leads to these deficits. Here the DD phenotype is evidenced by the reduced body length and head size. Since the heterozygous fish are not lethal, other phenotypes should be measured in these adult fish, which will strengthen the conclusion of this manuscript.

*As outlined in the response to Reviewer 1’s point 1 above, (1) *spout1* homozygous mutants showed significant differences of large effect compared to WT in eight assays, whereas heterozygous mutants showed a significant difference of modest effect in a*

single assay (locomotion); and (2) zebrafish mutants are biallelic LOF compared to humans, who predominantly harbor missense/missense or in rare cases missense/LOF variants. Furthermore, all the haploinsufficiency (HI) predictors indicate SPOUT1 is not a HI gene (%HI = 48.31; pLI = 0; and LOEUF = 0.98), and the gnomAD database reports >250 heterozygous LOF alleles in SPOUT1 suggesting that HI is unlikely to result in neurodevelopmental phenotypes in humans. We respectfully disagree that further characterization of heterozygous adult spout1 mutants will add substantially to the overall message of the manuscript.

Reviewer #2 (Remarks to the Author):

The authors have appropriately addressed my comments, and the revised manuscript has markedly improved.

Reviewer #3 (Remarks to the Author):

The authors have responded appropriately to most of my comments, and for the points they could not answer, they give convincing reasons. The actual molecular mechanism of SPOUT1/CENP-32 in the centrosome is not yet clear, but I don't think it is essential for this article, therefore it should not hinder the publication. In conclusion, I am satisfied with the revisions and recommend publication of this manuscript in Nature Communications. Please see the minor comments below to improve the quality of the manuscript.

1. Regarding the significant differences in fish body length (Fig. 2C, Fig. S4E, and Fig. S9B), I do not disagree with the authors' interpretation that the effect size is modest given the sample size. However, although the authors mention that they have revised the main text accordingly, I was unable to find this comment. Perhaps it would be better to insert it on line 340 in the revised manuscript. Please check and consider where best to place this comment.

*We apologize for the confusion from multiple spout1 models. We in fact had the following statement from the previous version of the manuscript, which is located on lines 352-355 of the current version (in the middle of the third paragraph of “Zebrafish spout1/cenp-32 depletion causes neuroanatomical defects” result section): “Consistent with F0 crispant data (Figure S4A-F), spout1/cenp-32 stable mutants exhibited a significantly reduced head size (20% reduction, $p < 0.0001$ for spout1/cenp-32^{-/-} vs spout1/cenp-32^{+/+}); **with modest but significant reduction in body length (Table S4; Figure 2A-C).**”*

2. Lines 1014, 1026 and 1116: “HEPESHEPES” should be corrected to “HEPES”.

Corrected.